# Structure of the endocytic adaptor complex reveals the basis for efficient membrane anchoring during clathrin-mediated endocytosis

Javier Lizarrondo[1,8], David P. Klebl [2,8], Stephan Niebling[1], Marc Abella[3], Martin A. Schroer [1], Haydyn D. T. Mertens [1], Katharina Veith[1], Roland Thuenauer[4], Dmitri I. Svergun [1], Michal Skruzny [3], Frank Sobott [5,6], Stephen P. Muench [2] & Maria M. Garcia-Alai [1,7 ✉]

During clathrin-mediated endocytosis, a complex and dynamic network of protein-membrane interactions cooperate to achieve membrane invagination. Throughout this process in yeast, endocytic coat adaptors, Sla2 and Ent1, must remain attached to the plasma membrane to transmit force from the actin cytoskeleton required for successful membrane invagination. Here, we present a cryo-EM structure of a 16-mer complex of the ANTH and ENTH membrane-binding domains from Sla2 and Ent1 bound to $PIP_2$ that constitutes the anchor to the plasma membrane. Detailed in vitro and in vivo mutagenesis of the complex interfaces delineate the key interactions for complex formation and deficient cell growth phenotypes demonstrate its biological relevance. A hetero-tetrameric unit binds $PIP_2$ molecules at the ANTH-ENTH interfaces and can form larger assemblies to contribute to membrane remodeling. Finally, a time-resolved small-angle X-ray scattering study of the interaction of these adaptor domains in vitro suggests that ANTH and ENTH domains have evolved to achieve a fast subsecond timescale assembly in the presence of $PIP_2$ and do not require further proteins to form a stable complex. Together, these findings provide a molecular understanding of an essential piece in the molecular puzzle of clathrin-coated endocytic sites.

[1] European Molecular Biology Laboratory, Hamburg Outstation, Hamburg, Germany. [2] School of Biomedical Sciences, Faculty of Biological Sciences and Astbury Centre for Structural and Molecular Biology, University of Leeds, Leeds, UK. [3] Department of Systems and Synthetic Microbiology, Max Planck Institute for Terrestrial Microbiology and LOEWE Center for Synthetic Microbiology (SYNMIKRO), Marburg, Germany. [4] Technology Platform Microscopy and Image Analysis, Heinrich Pette Institute, Leibniz Institute for Experimental Virology, Hamburg, Germany. [5] School of Molecular and Cellular Biology, Faculty of Biological Sciences and Astbury Centre for Structural and Molecular Biology, University of Leeds, Leeds, UK. [6] Department of Chemistry, Biomolecular and Analytical Mass Spectrometry group, University of Antwerp, Antwerp, Belgium. [7] Centre for Structural Systems Biology, Hamburg, Germany. [8] These authors contributed equally: Javier Lizarrondo, David P. Klebl. ✉email: garcia@embl-hamburg.de

Clathrin-mediated endocytosis (CME) is an essential cellular process that facilitates the internalization of external material, integral membrane proteins and lipids into eukaryotic cells. It is involved in many fundamental cellular processes, including nutrient uptake, cell signaling, cell adhesion and polarity, control of plasma membrane homeostasis, and synaptic vesicle recycling. CME also plays an important role in viral infection[1–3]. Clathrin mediates endocytosis through the formation of a protein coat around the endocytic site but it does not interact directly with the plasma membrane. Instead, adaptor proteins such as the AP-2 complex, proteins of the clathrin assembly lymphoid myeloid leukemia (CALM) family and epsins, connect the clathrin coat with the plasma membrane[4–8]. The high-resolution structures of polymerized clathrin coats have provided useful insights into the dynamic endocytic scaffold as a platform for the recruitment of other endocytic proteins and cargo at the membranes of clathrin-coated sites (CCS)[9–12].

In budding yeast, the endocytic process is well understood, and its timeline and components (with more than 60 proteins involved) have been described in genetic and microscopy studies[2,13–15]. Importantly, though clathrin is present in yeast endocytic sites, it is not an absolute prerequisite for endocytosis[16]. In addition, correlative light and electron microscopy (CLEM) has revealed that clathrin does not shape the membrane during invagination, but is instead required to determine the correct size and regularity of the endocytic vesicles[17]. Recent studies suggest that the adaptor proteins epsin Ent1 and Hip1R homolog Sla2 play a critical role in forming the protein coat at yeast endocytic sites[18].

Adaptors Ent1 and Sla2 co-assemble at the plasma membrane in a phosphatidylinositol 4,5-bisphosphate (PIP$_2$)-dependent manner through their epsin N-terminal homology (ENTH) and AP180 N-terminal homology (ANTH) domains, respectively[5,6,18,19]. These adaptors have the topology of elongated knot and string proteins, with globular domains connected by coiled-coil or intrinsically disordered regions (IDRs)[20–25]. They connect the plasma membrane and the actin cytoskeleton (through their C-terminal actin-binding domains), which allows them to gather the forces provided by actin for successful membrane invagination[18,21,22]. It is known that the PIP$_2$ local concentration is increased in the plasma membrane at the initial stages of endocytosis promoting the binding of adaptors and mediating the interface of the ANTH-ENTH interaction (from now on abbreviated AENTH) of Sla2 and Ent1[19,26–30]. Yet, the structural role of PIP$_2$ during membrane remodeling is still under debate, and our understanding of the nature and dynamics of membrane phospholipids and their associations with endocytic proteins during endocytosis remains limited. It is also intriguing how the weak protein-membrane interactions so far described can hold the plasma membrane during its remodeling without detaching from the membrane[3,5,6]. Another question involving endocytic adaptors is whether this remodeling is stabilized by multimeric complex protein scaffolds in addition to hydrophobic insertions of epsin and AP180 amphipathic α0 helices[5,31–35]. For all these questions the knowledge of the mechanism of ANTH and ENTH recruitment to the PIP$_2$-enriched membrane and the detailed structure of AENTH assembly is of high importance.

To understand the spatial and temporal regulation of the AENTH assembly, we determined the structure of a 16-mer complex of yeast Sla2 ANTH and Ent1 ENTH domains (formed by 8 ANTH and 8 ENTH subunits) bound to PIP$_2$ using single-particle cryo-EM and performed time-resolved small-angle X-ray scattering (SAXS) of the complex formation in solution. The structure unveils a hetero-tetrameric assembly of ANTH and ENTH domains as the building unit for larger assemblies. In the AENTH tetramer, the heterodimeric complementary interfaces are mediated by different PIP$_2$ molecules clamped between the different domains. Structure-based mutations of these interfaces lead to unstable complexes and growth deficiency phenotypes in vivo. We also show that the described ANTH-ENTH interaction is finely tuned as our kinetic studies reveal fast and efficient self-assembly. In vitro, assembly occurs on the sub-second scale and does not require an extra adaptor protein to be present. Finally, we show that the disassembly of the complex is a reversible process that could be regulated by decreasing local concentrations of any of the partners involved. Altogether, our findings provide a molecular understanding of key adaptor protein structure during endocytic vesicle formation and demonstrate that the mechanism of assembly is not based on weak lipid-protein interactions, but on lipid-mediated oligomeric states.

## Results

**Cryo-EM structure of the AENTH assembly.** The ANTH and ENTH domains from Sla2 and Ent1 (33.2 and 18.9 kDa, respectively, see Table 1) form discrete protein-lipid complexes in the presence of PIP$_2$ in solution[18,19,36]. Biophysical characterization of the in vitro reconstituted AENTH complexes by Dynamic Light Scattering (DLS) revealed the absence of large aggregates (Supplementary Fig. 1), which enabled single-particle cryo-EM structure determination of a 16-mer complex of ANTH-ENTH (A$_8$E$_8$) with PIP$_2$ (Fig. 1a, b, Supplementary Figs. 2 and 3). During particle classification in cryo-EM processing, heterogeneous oligomeric

**Table 1 Experimental and theoretical masses for ANTH, ENTH, PIP$_2$, and adducts/complexes observed by native MS.**

| Species | Ligand or adduct | Theoretical mass (Da) | Experimental mass (Da) | Main charge state | FWHM[a] |
|---|---|---|---|---|---|
| - | diC8-PI(4,5)P$_2$ | 747.5 | – | – | – |
| ENTH | - | 18847.9 | 18847.8 | 8+[b] | 1.2 |
| | +Na | 18870.9 | 18871.4 | 8+[b] | 1.1 |
| | +PIP$_2$ | 19594.4 | 19593.2 | 8+[b] | 1.1 |
| ANTH | - | 33210.0 | 33209.6 | 11+ | 1.2 |
| | +Na | 33233.0 | 33230.3 | 11+ | 1.3 |
| | ANTH + PIP$_2$ | 33956.5 | 33959.4 | 11+ | 1.3 |
| ANTH$_8$:ENTH$_8$ (wildtype) | +22 PIP$_2$ | 433181 | 432887 | 41+ | 9.7 |
| | +23 PIP$_2$ | 433892 | 433633 | 41+ | 10.5 |
| | +24 PIP$_2$ | 434627 | 434380 | 41+ | 9.2 |
| ANTH$_6$:ENTH$_6$ (ANTH R3A-I4-D37R-H38A) | +17 PIP$_2$ | 324125 | 324382 | 36+ | 8.5 |
| | +18 PIP$_2$ | 324872 | 325141 | 36+ | 10.7 |
| | +19 PIP$_2$ | 325618 | 325902 | 36+ | 12.2 |

[a]FWHM (full width at half maximum) is given for the predominant charge state peak to indicate the experimental error.
[b]Sensitivity for the Orbitrap UHMR instrument used here is reduced at low m/z (< 2500), so higher charge states of free ENTH may be attenuated.

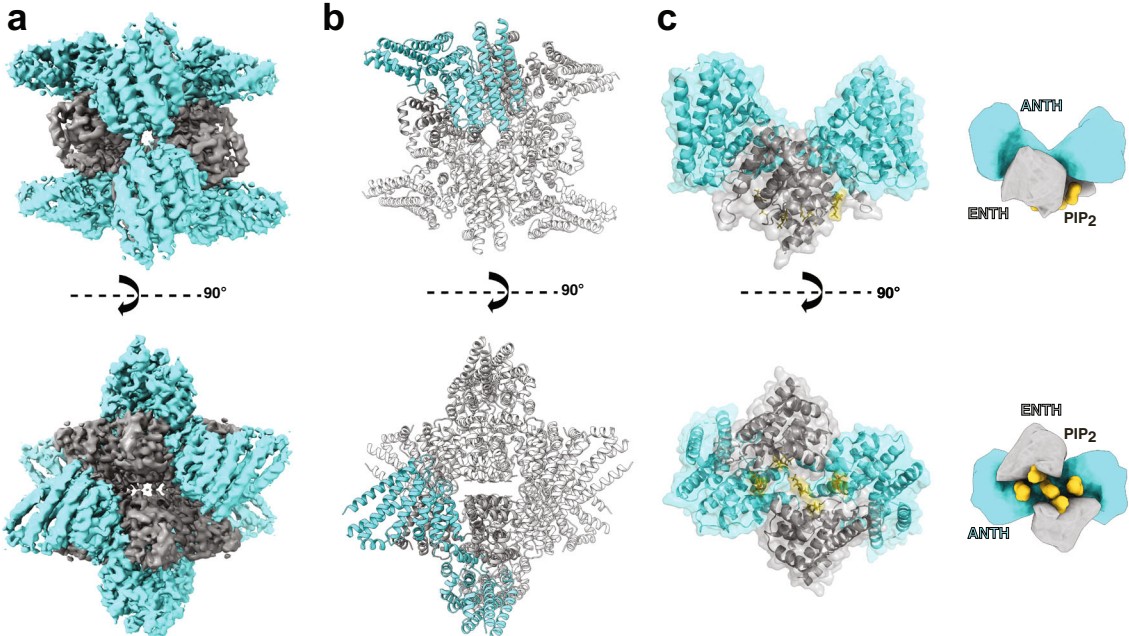

**Fig. 1 Cryo-EM structure of a 16-mer AENTH complex. a** Cryo-EM reconstruction obtained for the 16-mer complex formed by 8 ANTH and 8 ENTH units ($A_8E_8$). The density corresponding to ANTH and ENTH is colored correspondingly in cyan and gray. **b** Model of the $A_8E_8$ complex built based on the EM density shown in cartoon representation. One tetramer is colored with the ANTH domains in cyan and ENTH domains in gray (see Supplementary Fig. 4 for symmetry axes). **c** The building unit is the AENTH tetramer ($A_2E_2$) shown in cartoon and surface representation, with the ENTH domains in gray and the ANTH domains in cyan. The $PIP_2$ polar heads bound to the tetramer are shown in stick representation (gold). A schematic of the tetramer is shown next to the structure.

states were detected (Supplementary Fig. 2). A comparison of the theoretical scattering profile obtained for the 16-mer structure with SAXS curves for the AENTH complex samples also suggested heterogeneity in solution. To address this, SAXS curves were fitted using OLIGOMER[37] revealing an equilibrium of monomers and complexes in solution (Supplementary Fig. 1).

Focusing on the most abundant complex (16-mer), the structure reveals an assembly of 8 ANTH and 8 ENTH units arranged into four tetramers, each tetramer consisting of two molecules of ANTH and two of ENTH ($A_2E_2$) (Fig. 1c and Supplementary Fig. 4). The final reconstruction was obtained imposing D2 symmetry (Fig. 1b and Supplementary Fig. 2). Processing without applying any symmetry produced a similar structure but to lower resolution. The global resolution of the final EM map is 3.9 Å (gold-standard Fourier shell correlation threshold 0.143) with analysis of the local resolution showing higher resolution towards the core of the map, reaching 3.7 Å (Supplementary Fig. 3). Representative images of different regions of the map where the model was built are shown in Supplementary Fig. 5. The cryo-EM map shows that each $A_2E_2$ tetramer contains three distinct lipid-binding sites, harboring 5 lipids per $A_2E_2$ (Fig. 2) with a total of 20 $PIP_2$ lipids resolved in the entire 16-mer complex. The first binding site is located close to the ANTH KRKH motif, previously reported as a $PIP_2$ binding site, involving K24, K26, and H27 and in close proximity to K14[5]. Density present at this site could be assigned to the polar head of $PIP_2$ used for sample preparation (di-$C_8$-PI(4,5)$P_2$). Interestingly, the polar head of $PIP_2$ seems to also interact with residues K66 and K68 on the ENTH domain adjacent to this site (Fig. 2a). There are 8 $PIP_2$ molecules shared in this way between the ANTH and ENTH domains in our 16-mer structure, one per AENTH interface. In addition, density was present in the previously described lipid-binding pocket of the ENTH domain[6]. This binding site involves residues K3 and K10 on the ENTH α0 helix, the amphipathic helix involved in membrane bending, and

residues R24, N29, K61, and R62 in proximity to a $PIP_2$ molecule (Fig. 2b). Each ENTH domain contains one $PIP_2$ in its binding pocket, adding 8 $PIP_2$ bound to the 16-mer structure. Next to the ENTH $PIP_2$ binding site, there is another $PIP_2$ molecule (one per tetramer) located in the space between the two α0 helices of the ENTH domains in between their binding sites. In this case, the polar head of $PIP_2$ is coordinated by residues K10 and K14 of each of the two ENTH domains, establishing a total of 4 interactions that coordinate $PIP_2$ in this interface (Fig. 2c). Note that K10 coordinates both the $PIP_2$ molecules bound to the ENTH binding pocket and those shared between the two ENTH domains in the tetramer. Each tetramer contains one molecule of $PIP_2$ between the two ENTH domains adding a further four in the structure. In summary, our 16-mer structure shows 20 $PIP_2$ molecules, all of them clustered close to the core of the structure.

**The anchor unit of AENTH assembly is an hetero-tetramer.** Within each AENTH tetramer, the interactions between the ANTH and ENTH domains are established through two main hetero-dimerization interfaces and one homo-dimerization ANTH-ANTH interface. Thus, each AENTH tetramer contains a total of 5 interfaces (Fig. 3). Previous mutagenesis work on these domains described ENTH T104 and ANTH R29 as important residues for the functionality of Ent1 and Sla2 proteins in yeast[18,22]. These two residues were selected as the center of an ANTH-ENTH hetero-dimer fitted into a 13.6 Å resolution EM map of the domains' interaction on GUVs arranged in tubular structures[18]. Our structure shows that ENTH T104 and ANTH R29 are located in one of the observed interfaces, defined as "ANTH-ENTH interface 1" (Fig. 4a). Other residues present in this interface are ENTH Y100, E107, and ANTH R25, W36. ANTH R25 is in close contact with ENTH E107 and ENTH Y100 is oriented parallel to ANTH W36, establishing a stacking interaction by the coordination of their aromatic rings (Fig. 4a). To test

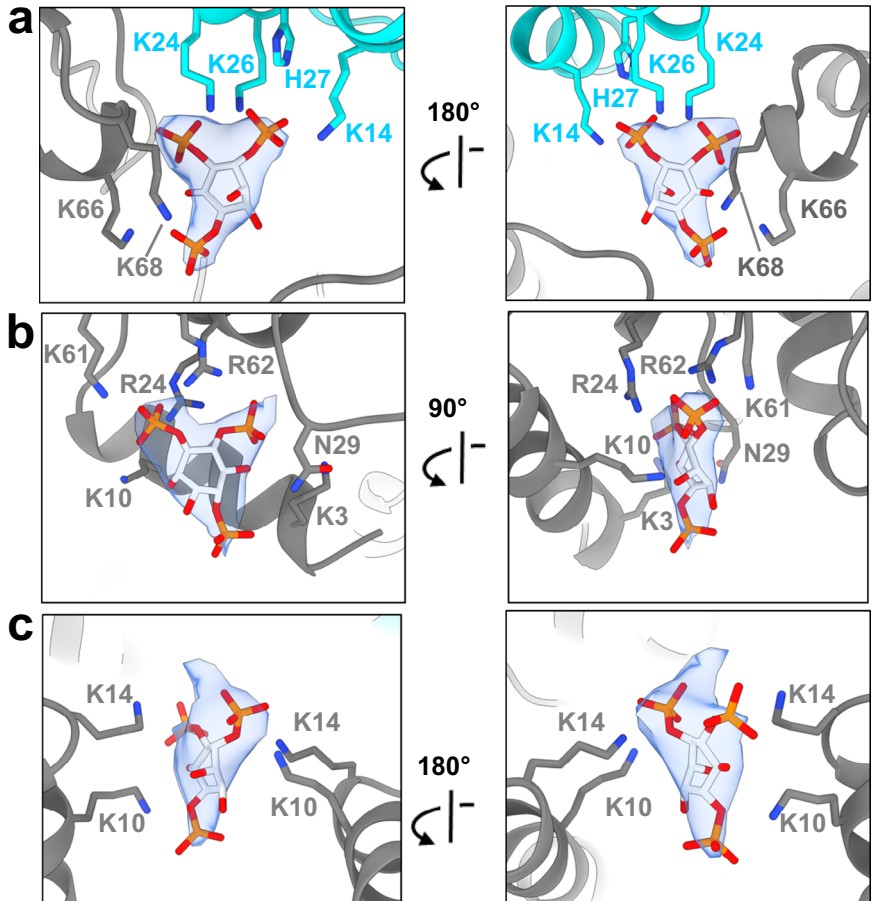

**Fig. 2 PIP$_2$ binding sites in the AENTH tetramer. a** PIP$_2$ binding site shared by the ANTH and ENTH domains. **b** PIP$_2$ binding site within the ENTH domain. **c** PIP$_2$ binding site in the interface between two ENTH domains. The residues involved in PIP$_2$ binding are shown in sticks and colored blue for nitrogen and gray and cyan for carbon in ENTH and ANTH, respectively. The polar head of the PIP$_2$ is shown in stick format and colored red, orange, and light gray for oxygen, phosphate, and carbon, respectively with the corresponding density shown as a transparent surface.

the relevance of this interface for complex formation, point mutations were introduced in residues in the ANTH and ENTH domains present in this interface, and the stability of recombinant proteins was analyzed by nanoDSF (Supplementary Fig. 6a).

AENTH complexes were assembled in the presence of PIP$_2$ in vitro (as done for the cryo-EM sample preparation) using mutants for one of the domains and its wild-type counterpart to assess the impact on the complex assembly by nanoDSF. Most mutant complexes showed a lower aggregation temperature when compared with AENTH wild-type (wt) (Supplementary Fig. 6d). ANTH and ENTH domains in the presence of PIP$_2$ form structures around 8 to 10 nm of average hydrodynamic radius in Dynamic Light Scattering (DLS) experiments. Mutation of these residues showed fewer particles corresponding to 16-mer assemblies in vitro when compared with AENTH wt complexes (Fig. 4d and Supplementary Table 1).

To explore with a greater level of detail the effect of these mutations on the assembly of AENTH complexes, non-denaturing electrospray ionization mass spectrometry (native MS) was used to analyze the assembly when mutant domains were used to reconstitute complexes (Fig. 4g). Native MS allowed the identification of charge state distributions corresponding to A$_8$E$_8$ complexes at high $m/z$, with 22–24 PIP$_2$ molecules bound (Supplementary Fig. 7 and Table 1). The ENTH Y100R mutant did not show 16-mer assemblies in vitro (Fig. 4g). In vivo, the growth-defect phenotype of ENTH Y100R mutant has been previously attributed to a possible role of Ent1 in binding GTP-

activating proteins for Cdc42, a critical regulator of cell polarity[35]. Here, we show that this residue plays a crucial role in the formation of the AENTH complex. In addition, a mutation in vivo of ANTH R25, R29, and W36 and ENTH Y100 and E107 showed strong or intermediate growth deficiency phenotypes in yeast, confirming an essential role of this interface for proper endocytic function (Fig. 4h, i). ENTH F108A also introduced a growth defect phenotype in vivo (Fig. 4i), but our in vitro data showed that the stability of the protein is compromised by this mutation (Supplementary Fig. 6a). A similar scenario was found for ANTH Y247 and L248, conserved specifically in ANTH domains of the Sla2 family, whose deletion caused an endocytosis-linked growth defect in yeast, previously attributed to a possible interface in the protein complex with epsin[19]. Our structure shows that these residues are not involved in any interface of the assembly, but instead that their deletion leads to an unstable protein in vitro suggesting that they are crucial for the folding of ANTH (Supplementary Fig. 8).

A second interface in the AENTH tetramer involves residues K10, K13, and K14 of ANTH (Fig. 4b and Supplementary Fig. 6b). ANTH K14 is involved in the interaction with PIP$_2$, indicating synergy in PIP$_2$ binding and function of this ANTH-ENTH interface 2. After testing that mutations of the lysine residues in this interface did not affect the stability of the ANTH domain (see "Methods" section and Supplementary Fig. 6b), we observed that these mutants display AENTH complexes with a lower aggregation temperature ($T_{agg}$) by nanoDSF, indicating that

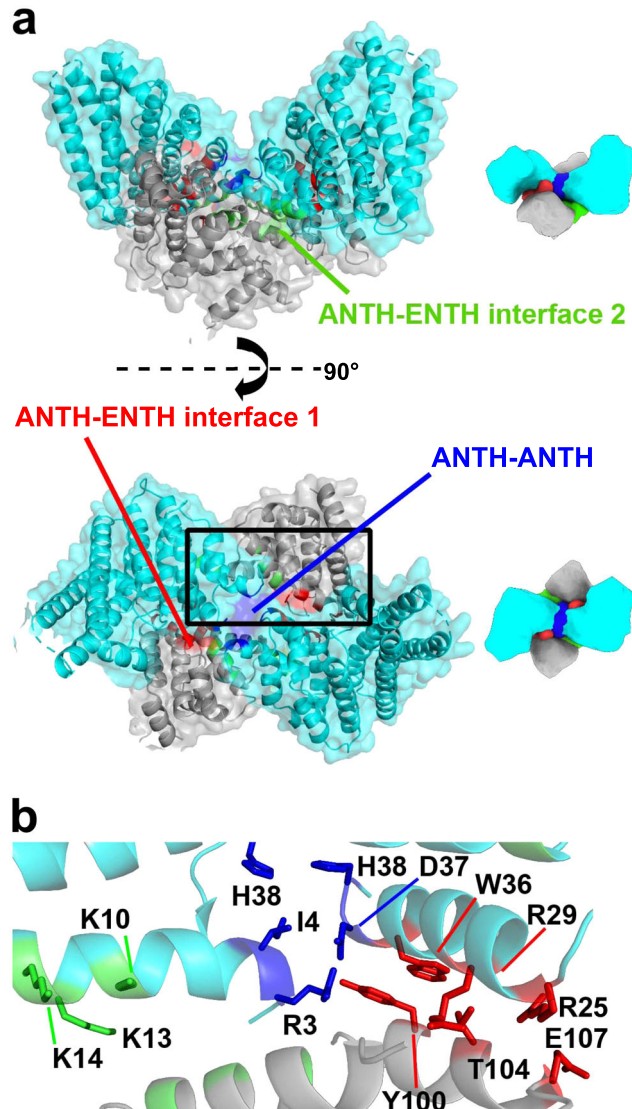

**Fig. 3 The AENTH tetramer interfaces. a** Surface representation of the $A_2E_2$ tetramer, with ANTH subunits in cyan and ENTH subunits in gray. The different interfaces are colored: ANTH-ENTH interface 1 in red, ANTH-ENTH interface 2 in green and ANTH-ANTH interface in blue. **b** Cartoon representation of the tetramer area within the black rectangle in a showing the different interfacial residues as sticks (same color code as in a).

the stability of these complexes is partially compromised (Supplementary Fig. 6e). Apart from this destabilization effect, some mutants also showed a clear effect on complex assembly reported by DLS (Fig. 4e and Supplementary Table 1) when compared with AENTH wt complex. Specifically, triple mutant ANTH K10A/K13A/K14A showed a smaller proportion of 16-mer assemblies in solution, while mutant ANTH K10D/K13D/K14D completely abolished complex formation (Fig. 4e).

Moreover, ANTH K10D/K13D displays a large destabilization of the complex observed by nanoDSF (Supplementary Fig. 6e) and only a weak native MS signal corresponding to the 16-mer assembly when compared to the wild-type (Fig. 4g). Together, these results indicate that K10 and K13 residues are important for oligomerization. Beyond that, ANTH K10D/K13D/K14D completely abolished complex formation, and no signal for larger assemblies other than monomeric species was observed by native MS in agreement with the DLS data (Fig. 4g). In vivo, mutations of these residues caused growth defect phenotypes in yeast strains

lacking wild-type Sla2 (Fig. 4j and Supplementary Fig. 9b). Mutation of ANTH K10/K13 to alanine caused an intermediate growth defect phenotype which was enhanced when these residues were mutated to glutamic acid. The tripe mutant K10/K13/K14 present in this interface caused a severe growth defect phenotype when mutated to alanine, further enhanced when replaced by glutamic acid. Furthermore, each of the three lysine residues has an important role in complex formation, as individual mutations of these lysines also caused growth defect phenotypes in vivo (Supplementary Fig. 9b). Altogether, these results indicate that lysines of the ANTH-ENTH interface 2 are essential for the assembly of the tetrameric lipid-binding AENTH unit and for the AENTH function in vivo. On the ENTH domain, E54A/D57A/D60A showed the largest destabilization in the nanoDSF data (Supplementary Fig. 6e). However, mutation of these residues did not show any complex disruption by DLS in vitro nor introduced a growth defect in yeast cells, most likely ruling out a major role in the AENTH assembly and function (Supplementary Fig. 9a).

Finally, ANTH-ANTH interface mutants (Supplementary Fig. 6c) did not show a large destabilization effect over the complex in vitro with the exception of ANTH R3A (Supplementary Fig. 6f) which also generated a larger amount of monomeric species upon complex formation when compared to AENTH wild-type by DLS (Fig. 4f and Supplementary Table 1). Interestingly, native MS for the R3A/I4A/D37R/H38A mutant showed a shift in the signal of the complexes obtained to lower m/z, corresponding to 12-mers (Fig. 4g and Table 1), in agreement with the DLS data. Assemblies of 12-mers have been previously reported as lower abundance species[19,36] and are formed by 6 ANTH and 6 ENTH molecules (termed $A_6E_6$). However, mutation of the ANTH-ANTH interface did not cause growth defect phenotype in vivo (Supplementary Fig. 9c).

ENTH α0 helix mutant F5A/L12A/V13A also produced a native MS spectrum with signal only corresponding to the 12-mer assembly (Supplementary Fig. 10a). Single-particle cryo-EM data for AENTH complexes formed with ENTH F5A/L12A/V13A showed a structure distinct from the $A_8E_8$ assembly (Supplementary Fig. 10 and Supplementary Table 2). The density was unambiguously assigned to three $A_2E_2$ tetramers arranged around a central core of PIP$_2$ (Supplementary Fig. 10b and Supplementary Fig. 10c). The position of the ANTH and ENTH domains within the $A_2E_2$ tetramer is remarkably similar to the $A_8E_8$ complex, indicating that mutation of hydrophobic residues on the amphipathic helix does not disrupt the ability of the ENTH domains to assemble into the functional hetero-tetramer with the ANTH domains. All these observations support the hypothesis that the building units of AENTH complexes are hetero-tetramers that can form different higher-order assemblies.

**AENTH tetrameric assemblies are relevant for membrane remodeling.** In our 16-mer assembly, all the PIP$_2$ polar head groups are located close to the core of the structure (Fig. 5a). Since samples were prepared above the critical micelle concentration (CMC) of di-C$_8$-PI(4,5)P$_2$[38,39], the position of the polar groups indicates that the core of the map is composed of a PIP$_2$ micelle around which the ANTH and ENTH domains assemble. However, there is no density present at the core of the map, probably due to the flexibility of the hydrocarbon tails of the lipids preventing them from being resolved. PIP$_2$ micelles were measured during our DLS experiments and showed a hydrodynamic radius between 0.9 and 1.4 nm. In addition, all ENTH domains arrange themselves around the micelle displaying their α0 helices into its hydrophobic core (Fig. 5b). All this suggests that our 16-mer AENTH complex is assembled on a minimal

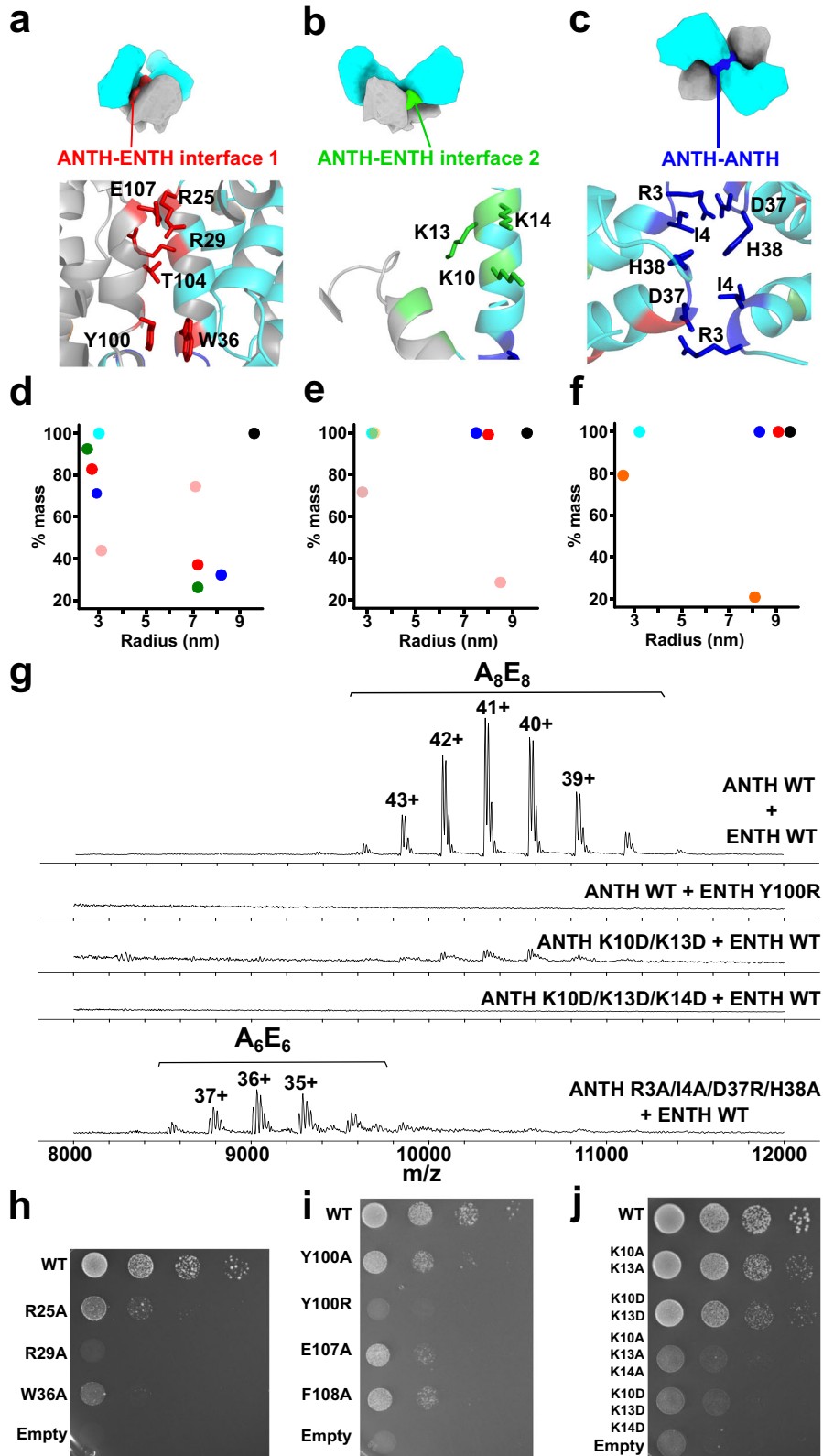

micelle made of di-$C_8$-PI(4,5)$P_2$ molecules. In addition to the 20 PIP$_2$ lipids described in the entire 16-mer complex, up to 24 PIP$_2$ molecules were detected by native MS indicating the presence of four additional bound lipid molecules that could not be resolved in our structure (Supplementary Fig. 7c).

Previous work showed that the ANTH and ENTH domains together tubulate GUVs and coat them with regular helical

assemblies, as determined by cryo-EM to 13.6 Å resolution[18]. Our 16-mer structure solved by single-particle cryo-EM suggests that an AENTH tetramer could be the building unit of larger clusters and assemblies. Therefore, we used our AENTH tetramer for flexible fitting into the electron density map of ANTH-ENTH coat on GUVs (Fig. 5b–d). Overall, the fit agrees with the previous assignment of the domains to the larger and smaller

**Fig. 4 Different interfaces present in the AENTH tetramer. a–c** Schematic representation of the AENTH model showing the different interfaces present, designated as ANTH-ENTH interface 1 (**a**), ANTH-ENTH interface 2 (**b**), and ANTH-ANTH interface (**c**). In the lower panels, the residues involved in each of the interfaces are shown in stick representation and colored using the color scheme for each of the interfaces. (**d–f**) Intensity mass plot from DLS data showing the hydrodynamic radius of particles in solution for wild type AENTH (black) and different mutants. As a reference, the ANTH domain is shown in cyan (see Supplementary Table 1 for DLS parameters). **d** ANTH-ENTH interface 1 showing mutations ANTH Y100R (red) ANTH R25A (blue), ANTH R29A (green) and ENTH E107A (pink); (**e**) ANTH-ENTH interface 2 with mutations ANTH K10A/K13A/K14A (pink) ANTH K10D/K13D/K14D (yellow), ENTH E54A/D57A/D60A (blue) and ANTH Q9A/K10A (red). (**f**) ANTH-ANTH interface with mutations ANTH R3A (orange), ANTH R3A/I4A/D37R/H38A (blue), and ANTH R3A/I4A/D37R (red). **g** Summary of the native mass spectrometry results obtained for the mutants of the tetramer interfaces. **h–j** Growth defects of mutants of ANTH (**h**) and ENTH (**i**) ANTH-ENTH interface 1 and ANTH-ENTH interface 2. (**j**) Interface mutants were expressed after depletion or deletion of endogenous Ent1 and Sla2 proteins, respectively. Cell growth was analyzed after plating a 10-fold serial dilution of cells on SD-Ura plates and incubation for 3 days at 37 °C.

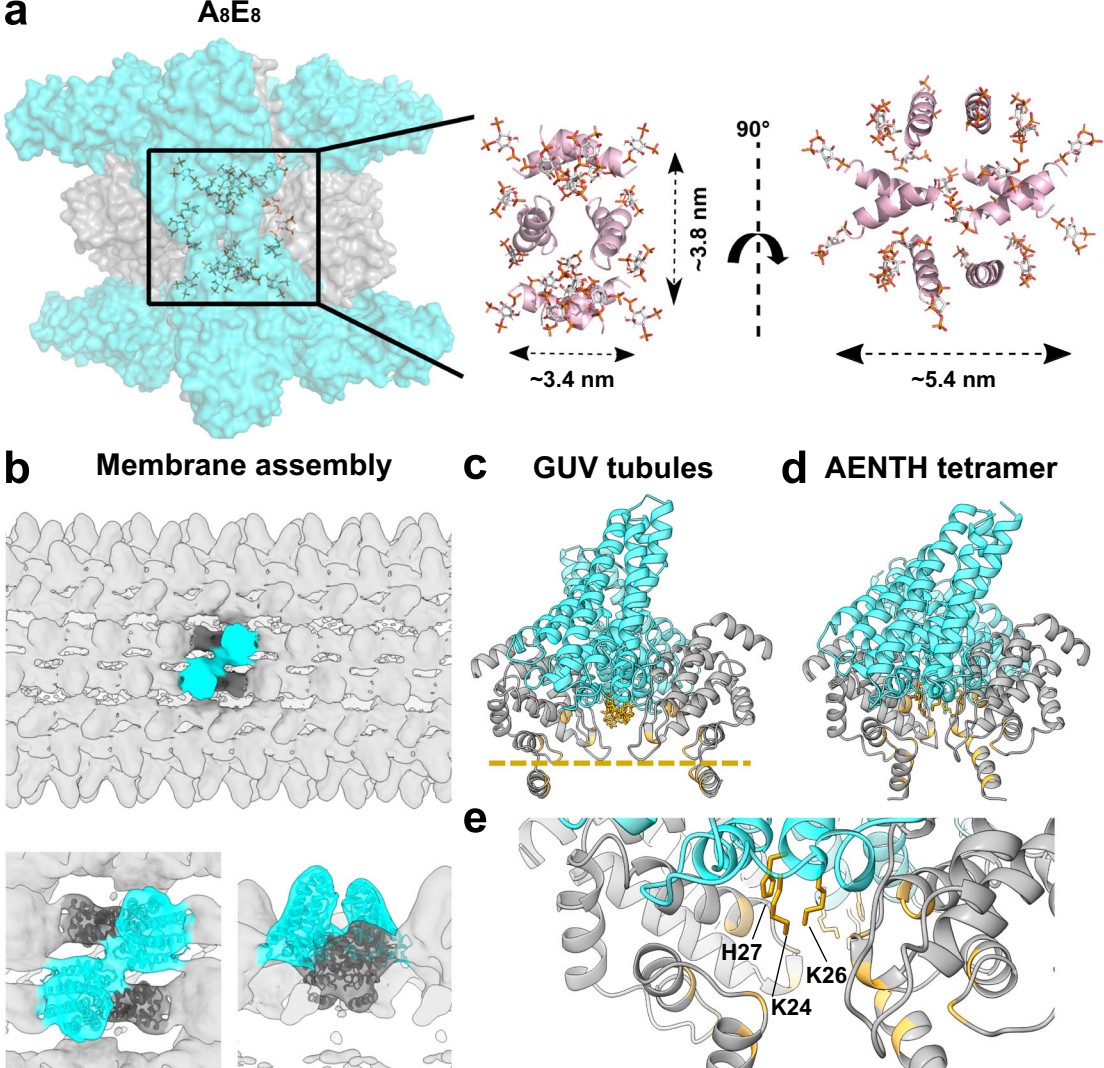

**Fig. 5 Protein-lipid interactions of the AENTH 16-mer complex structure and of the ANTH-ENTH assembly on GUVs. a**, left, All the polar heads from the PIP$_2$ placed in the structure (shown in sticks) are contained in the region near to the core of the structure (shown in surface representation), indicating the presence of a PIP$_2$ micelle in the center of the map. Right, only the α0 helices from ENTH subunits are shown in cartoon representation (in pink) together with the PIP$_2$ shown in stick (colored by atom), all of them are pointing towards the interior of the structure. **b–e** Flexible fitting of the tetrameric model to the previously obtained cryo-EM map of ANTH-ENTH coat on GUV tubules[18]. **b** Overview of the GUVs tubular coat structure with the tetramer fitting inside the lobes of the structure (EM-DB entry: EM-2896). Bottom, close-up view of the tetramer fitted into the EM density of the GUV structure. **c** AENTH tetramer structure fitted to the GUVs coat structure. The regions involved in PIP$_2$ contacts are colored in gold in the cartoon representation. The bilayer plane is indicated as a gold dashed line. **d** AENTH tetramer in the 16-mer ANTH-ENTH complex, with the lipid-binding regions highlighted in gold in the cartoon representation. **e** Close-up for the ANTH lysine patch residues involved in the coordination of the polar head of PIP$_2$ lipid.

densities present on the surface of the tubules for the ANTH and ENTH domains, respectively. It also places the ENTH α0 helixes pointing towards the core of the tubules, consistent with the membrane bending mechanism of the ENTH domain by insertion of the α0 helix and displacement of the lipids in the inner layer of the plasma membrane[31–33] (Fig. 5b). Similarly, the ANTH domains have their lysine patches available to bind the polar heads of the PIP$_2$ at the membrane, in agreement with its lipid-binding mechanism[5] (Fig. 5d, e). In spite of the close similarity between the membrane-bound model from the previous helical reconstruction and the single-particle cryo-EM structure presented here, the distance between the two ENTH domains of the tetramer seems to be much larger in the membrane-bound model (see supplementary video 1). Therefore, the membrane-bound tetramer is in a more "open" conformation compared to the 16mer structure solved with PIP$_2$ (Fig. 5c, d). Mechanistically, this difference between the membrane-bound and the soluble PIP$_2$ structure could potentially reflect the tetramer exerting force on the membrane.

To address the relevance of the AENTH tetramer in membrane binding and remodeling, we studied the effects of protein interface mutations in in-vitro model membranes. We used giant unilamellar vesicles (GUVs) containing PIP$_2$ and mixed them with wt ANTH-GFP or ENTH-GFP fusions and interface mutants, respectively, and observed them by confocal fluorescence microscopy. The addition of wild-type ANTH and ENTH domains led to membrane remodeling and resulted in elongated hairy structures protruding from GUVs, while the individual domains did not cause any major membrane re-shaping phenotype (Fig. 6a–f). Those interface mutants that were not able to assemble the A$_8$E$_8$ complex: ENTH Y100R, ANTH R29A, ANTH K10D/ K13D, and ANTH K10D/K13D/K14D did also not introduce membrane remodeling (Fig. 6i, k, l, m), while the mutant E54A/D57A/D60A, which still enabled complex formation, elicited similar hairy membrane structures as wt ENTH (Fig. 6h).

Higher curvature liposomes (LUVs) containing PIP$_2$ were prepared by extrusion and their interactions with ANTH and ENTH visualized by negative stain EM. The addition of ANTH

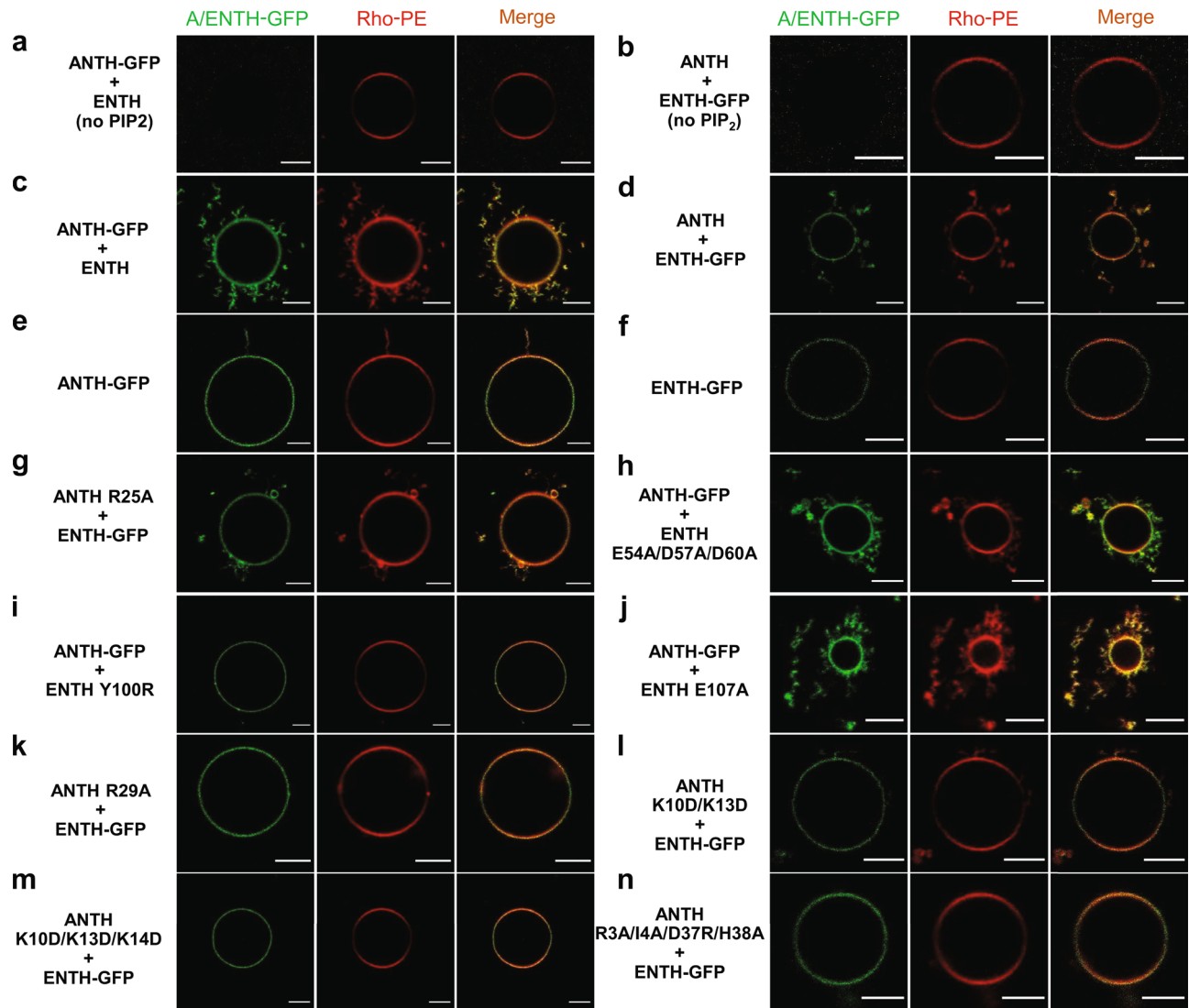

**Fig. 6 Effect of AENTH tetramer and mutants on GUVs by fluorescence microscopy. a, b** ANTH-GFP and ENTH-GFP do not bind GUVs without PIP$_2$. **c, d** ANTH-GFP or ENTH-GFP mixed with ENTH or ANTH, respectively, caused membrane deformation as seen by the numerous membrane protrusions around GUVs. **e, f** ANTH-GFP or ENTH-GFP did not cause a remodeling effect on the GUV membrane. **g–n** ANTH and ENTH mutants mixed with ENTH-GFP and ANTH-GFP, respectively, showed different membrane deformation capabilities. Each experiment was repeated independently 3 times with similar results. All scale bars are 5 μm.

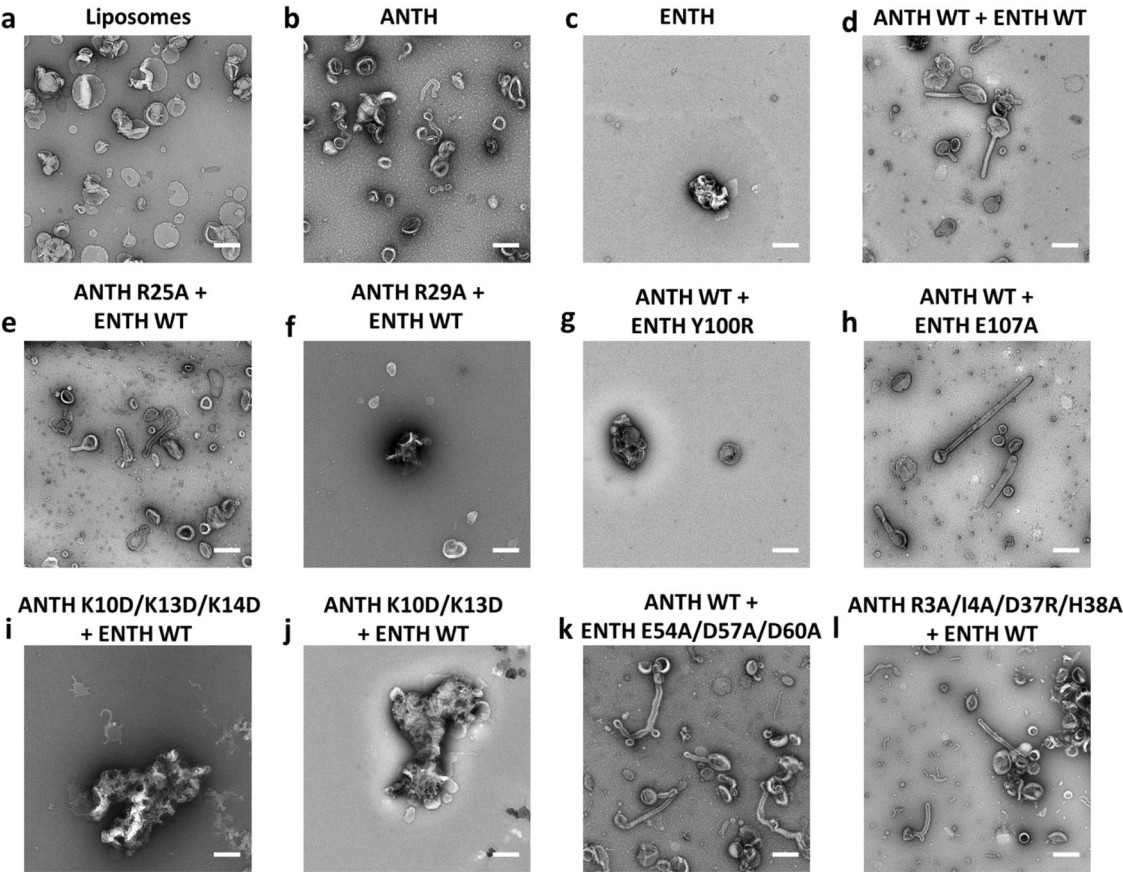

**Fig. 7 Effect of AENTH tetramer mutants on LUVs by the negative stain. a** LUVs observed by negative staining. **b** ANTH does not deform LUVs notably. **c**. ENTH causes LUVs to become aggregated and to adopt irregular shapes. **d** ANTH together with ENTH reshapes LUVs into tubular structures as previously reported[18]. **e–l** ANTH and ENTH mutants show different effects on LUVs. ANTH R25A (**e**), ENTH E54A/D57A/D60A (**k**) and ANTH R3A/I4A/D37A/H38A (**l**) mutants (mixed with its wild type partner domain) display an AENTH wt phenotype reshaping LUVs into tubular structures ANTH R29A (**f**) and ENTH Y100R (**g**) ANTH K10D/K13D/K14D (**i**) and ANTH K10D/K13D (**j**) mutants (mixed with its wild type partner domain) display an ENTH wt phenotype being unable to tubulate LUVs. Each experiment was repeated independently 3 times with similar results. All scale bars are 200 nm.

alone did not deform LUVs containing PIP$_2$, whereas ENTH caused their aggregation (Fig. 7b, c). Corroborating what was already observed for GUVs, the addition of wild-type ANTH and ENTH domains together caused tubulation of liposomes (Fig. 7d). Mutations on the "ANTH ENTH interface 1" had a strong effect on membrane remodeling with ANTH R29A and ENTH Y100R displaying aggregation of LUVs without causing tubulation (Fig. 7f, g). "ANTH ENTH interface 2" ANTH lysine mutants (K10D/K13D/K14D and K10D/K13D), incapable of oligomerization with ENTH, induced a strong aggregation of LUVs (Fig. 7i, m, j). Similar to what was observed for GUVs, ENTH E54A/D57A/D60A was able to tubulate LUVs, in agreement with our observations for complex assembly followed by DLS and native MS.

The ANTH R25A mutant was shown to partially impair complex assembly in vitro and exhibits an intermediate growth defective phenotype in vivo (Fig. 4d, h). However, the mixture of ANTH R25A and ENTH was sufficient to induce tubulation of both GUVs and LUVs (Fig. 6g and Fig. 7e). Similarly, ENTH E107A, which introduced a mild growth defect phenotype in vivo (Fig. 4i) did not show a clear effect on membrane remodeling in vitro, capable of tubulate GUVs and LUVs (see Fig. 6j and Fig. 7h). Finally, "ANTH-ANTH interface" mutant ANTH R3A/I4A/D37R/H38A, that introduced intermediate effects on complex destabilization (Fig. 4f–g and Supplementary Fig. 6f) and no growth defect phenotypes in vivo (Supplementary Fig. 9c) was capable of tubulating LUVs but not GUVs (Figs. 6n and 7l). In

the case of ANTH R3A/I4A/D37R/H38A, the typical 16-mer AENTH complex was not observed for this mutant, but a 12-mer assembly (Fig. 4g and Table 1) was sufficient to introduce membrane reshaping in vitro.

**Kinetics of the AENTH complex assembly.** In equilibrium, ENTH and ANTH form protein-lipid assemblies in the presence of PIP$_2$, whose major component is the 16-mer A$_8$E$_8$ AENTH complex determined by cryo-EM. Supplementary Fig. 1 shows the SAXS data from this equilibrium in solution at different concentrations. To study compatibility with cellular time scales of such assembly, we performed stopped-flow time-resolved SAXS (SF-TR-SAXS)[40] studies of the interaction upon fast mixing of its constituents. An SF device enabled rapid mixing of ANTH and ENTH domains followed by immediate exposure to X-ray radiation to monitor the scattering signal over time. We observed a fast change in the q-region between 0.02 to 0.6 nm$^{-1}$ ($q = 4\pi \sin\theta/\lambda$) of the buffer-subtracted SAXS curves over the first 400 ms (Fig. 8a). The radius of gyration R$_g$ extracted from the Guinier region of the curves for all time points showed a fast increase and then remained stable over time, indicating that the formation of larger oligomers takes place immediately after mixing (Fig. 8b). The singular value decomposition (SVD) analysis revealed three major components[40] (see "Methods" section and Supplementary Fig. 11). As the assembly was started from monomers and evolved into 16-mers, an ab initio program

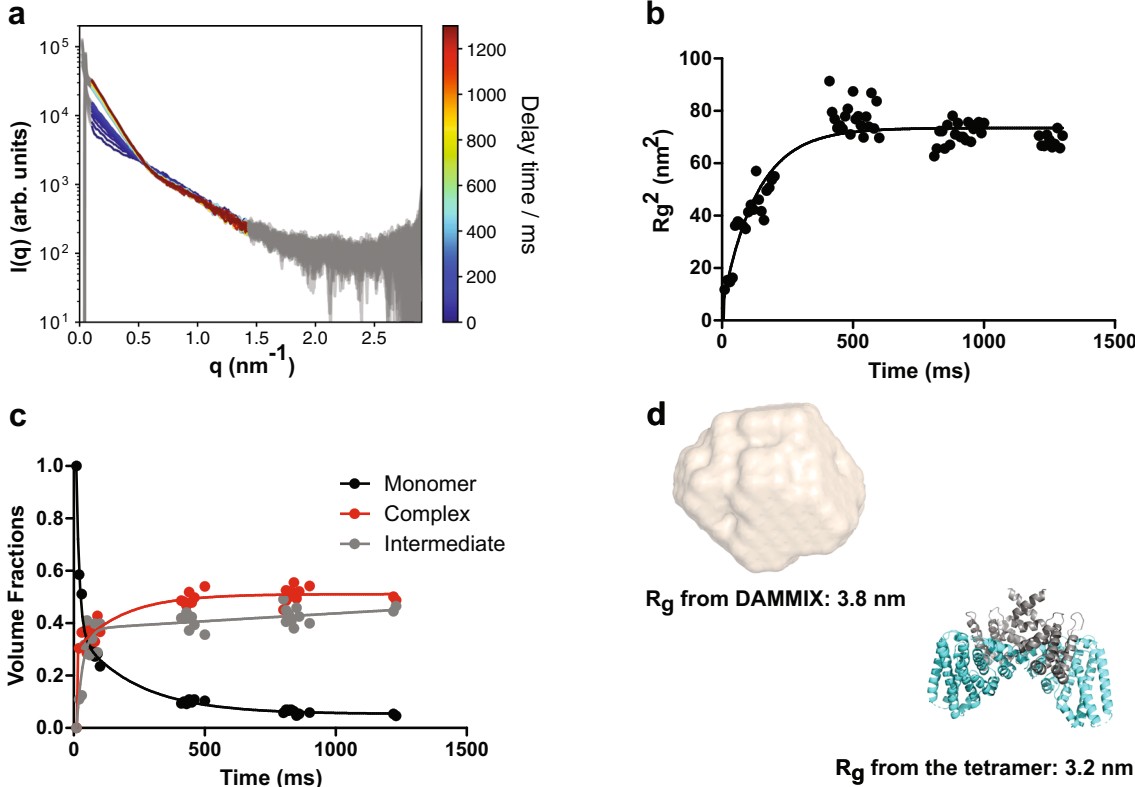

**Fig. 8 Time-resolved SAXS data demonstrate a fast assembly of ANTH and ENTH domains in solution. a** Buffer-corrected SAXS curves for the time-resolved measurements of the complex formation between ANTH and ENTH in presence of 200 μM $PIP_2$. For a better visibility, the data were smoothed with a Savitzky-Golay filter (51-point window, $2^{nd}$ order polynomial). Delay times are color-coded from 0 ms in blue to 1200 ms in dark red. The shape of the curve changes dramatically from the initial time points after 400 ms when the curve stabilizes. **b** Evolution of the $R_g^2$ data of the time-resolved SAXS data set over time. After 500 ms the $R_g$ is relatively constant in all curves. A single exponential fitting was performed and the half-time obtained was ~91 ms. **c** Volume fractions of DAMMIX obtained using the SAXS curves. The contribution of the unknown component is labeled as "Intermediate". The volume fractions were fitted to a bi-exponential function to obtain the time constants (fast constant ~8 ms, slow constant around ~150 ms). **d** DAMMIX ab initio model obtained for the intermediate component vs AENTH tetramer structure from the 16-mer cryoEM map solved. The dimensions of the tetramer and the DAMMIX ab initio model are rather similar, which could indicate that the unknown component of the SAXS data could be tetramers forming in a solution that later on assemble further into larger oligomeric states.

DAMMIX[41] was employed to determine the shape and fraction of the intermediate component of AENTH. The resulting monomer component (first curve) decreased over time, while the volume fraction of the complex (last curve of our dataset) together with the intermediate component, increased with time until around 400 ms, similar to the observed $R_g$ increase (Fig. 8c). The low-resolution model of the intermediate component showed similar dimensions to the AENTH tetramer (Fig. 8d). However, given the heterogeneity present in the solution, this structural model should be considered just as a representative average of the oligomeric states in the solution during the millisecond timescale. In conclusion, our SF-TR-SAXS data indicate that the association of the domains takes place very rapidly and even faster than the fast oligomerization occurring during endocytic coat assembly in vivo, where it happens in the range of seconds[1,2,42]. Interestingly, the TR-SAXS reveals a mixture of (largely) tetramers and 16-mers at the end of the time range probed (1.4 s). This result further corroborates the finding that much longer incubation times (minutes) are needed for the formation of (nearly) monodisperse 16-mer solution.

Finally, once endocytosis has been accomplished, the endocytic coat has to be disassembled and its components recycled. We performed biolayer interferometry (BLI) experiments to obtain information regarding the reversibility of complex formation. Our results indicate that complex formation is fully reversible

upon decreased concentrations of any of the two adaptor domains, as the signal decreases exponentially back to the baseline upon removal of their interaction partner (Supplementary Fig. 12 and Supplementary Table 3). This indicates that these domains are able to disassemble while the endocytic coat is being dismantled and the local concentration of one of their partner decreases.

## Discussion

Clathrin-mediated endocytosis requires the concerted action of many proteins to coordinate cargo recruitment, endocytic coat formation, and membrane bending. The early stages of the coat formation are essential for clathrin recruitment and proper assembly of the endocytic machinery for downstream actin polymerization and membrane invagination. We need to understand how the spatial and temporal regulation of CME functions to control relevant cellular processes involving adhesion complexes, signaling receptors, mechano-transducers, morphogen sensors, or polarity markers. Here, we determined the structure of a 16-mer assembly of the membrane-binding domains of Hip1R homolog Sla2 and epsin Ent1, ANTH and ENTH domains, respectively, with $PIP_2$ (8 ANTH:8 ENTH:20 $PIP_2$) using cryo-EM. We found that the ANTH and ENTH domains arrange into tetrameric complexes that can further assemble into larger

assemblies. The term di-octamer used in previous studies should therefore be re-interpreted as four tetramers[19,36]. The AENTH tetramer is also the building unit capable to contribute to membrane remodeling during endocytosis, as illustrated by the ANTH-ENTH helical assembly on GUVs previously reported[18] and corroborated by our experiments with AENTH tetramer interface mutants on GUVs and LUVs.

PIP$_2$ was found in three different binding sites, one in the ENTH domain, previously known from the crystal structure[5], one in the interface between the two ENTH domains of one tetramer, and one in the interface between the ANTH and ENTH domains, acting as a 'glue' for the oligomerization of both domains. The different binding sites present in the ENTH domain and especially shared binding sites between ANTH and ENTH domains agree with the positive cooperativity in the binding mechanism proposed before[18,19].

The endocytic mechanism involves a plethora of protein-protein and protein-lipid interactions, however, the strong affinity of these two domains for PIP$_2$, which is specific to the endocytic site, makes it different from most other interactions in the endocytic coat. These are usually of low affinity to provide the coat with the dynamic characteristics required for this finely tuned biological process. The structure of the A$_8$E$_8$ complex suggests that binding of the PIP$_2$ to the ENTH domain, causing refolding of its α0 helix, is not sufficient to tightly anchor the ENTH domain and therefore epsin to the plasma membrane. A low affinity of ENTH for the membrane could be deleterious for endocytosis in yeast or in cells with high membrane tension, where CME depends on forces provided by actin and transmitted over the Sla2-epsin linker[21,22,43]. To achieve high affinity for PIP$_2$-enriched membranes of endocytic sites, the ANTH domain must be present, sandwiching two PIP$_2$ molecules shared with the ENTH domain and increasing thus the avidity of the ANTH-ENTH tetrameric complex to the membrane. Our tetrameric structure suggests a mechanism of lipid locking as the anchor unit to the membrane that would enable force transmission coming from actin polymerization for invagination to take place. Its efficiency relies on sharing multiple PIP$_2$ molecules between the two adaptors.

Our data suggest that ANTH and ENTH domains have evolved to achieve a fast assembly in presence of PIP$_2$ and to do not require further proteins to form a stable complex. The whole endocytic process in yeast takes approximately 60 seconds[1], and adaptors such as Sla2 and Ent1 assemble quickly and efficiently in the plasma membrane during the coat formation to provide a scaffold for further assembly of the endocytic machinery. SAXS and BLI data highlight that this dynamic system can be quickly self-assembled and disassembled. The assembly takes place in the 100 ms scale and the intermediate component is on the range of dimensions of the tetramer. Together with our cryo-EM data, this supports a context in which different tetramers could be the building units for this dynamic lipid-protein anchoring system.

It is biologically relevant that the kinetics of the oligomerization of these domains in solution reveals a very fast sub-second scale assembly of these essential pieces of the endocytic coat. Failure to do so would result in hampered endocytosis. An efficient disassembly should be then achieved with the help of phosphatases, which at the end of the endocytic cycle remove PIP$_2$ from the endocytic vesicle favoring the equilibrium of the oligomerization towards the disassembly of the endocytic adaptor complexes[1,26].

In summary, ANTH and ENTH domains can assemble in a fast and coordinated way into their hetero-tetrameric functional unit, stabilized by shared PIP$_2$ molecules. These tetramers can give rise to different assemblies depending on the lipid environment and membrane shapes occurring during endocytosis. The synergy of

structural properties from ENTH α0 helices, combined with the lipid clamp of Sla2 ANTH domains makes the complex a better membrane anchor relying on shared PIP$_2$ molecules in several protein interfaces. Thus, the combination of these two elements transforms the monomeric low-affinity (micromolar $K_d$) interaction with phospholipids[3,5,6] into a nanomolar interaction[18,19] provided by a highly organized protein-lipid-protein complex. This multimeric membrane anchor is then essential for membrane invagination in organisms and cells where endocytosis is challenged by high membrane tension or turgor pressure.

## Methods

**Protein production and purification.** Recombinant yeast Sla2 ANTH and Ent1 ENTH domains were expressed in *E. coli* BL21 DE3 (Novagen) as GST-fusion proteins containing an N-terminal His-tag followed by a TEV (Hisx6-TEV) cleavage site. Flasks containing 800 ml cultures in LB media were grown at 37 °C until an optical density (OD$_{600}$) of 0.8. After induction with 0.5 mM Isopropyl β-d-1-thiogalactopyranoside (IPTG), the cultures were grown at 20 °C for 4 h and harvested by centrifugation (4000 × *g* for 30 min at 4 °C). The cell pellet was lysed by sonication in the presence of 1 mg/ml DNase in 50 mM Tris-HCl pH 7.5, 300 mM NaCl, 20 mM imidazole. Lysed cell extract was centrifuged (17,000×*g*, 45 min at 4 °C) and the supernatant containing His-tagged proteins were purified by nickel-nitrilotriacetic acid (Ni-NTA) purification (Qiagen). Protein was eluted in a final elution buffer of 20 mM Tris-HCl pH 8.0, 300 mM NaCl, 250 mM imidazole. Excess of TEV protease was added to the imidazole-eluted fractions for cleavage of the His$_6$-GST and His$_6$ tags. Digestion was performed by dialysis at 4 °C overnight against 5 L of 20 mM Tris-HCl pH 8.0, 250 mM NaCl, and 1 mM Dithiothreitol (DTT). To remove the tags, the dialyzed fractions were subjected to a second Ni-NTA and the flow-through was concentrated to 5 mg/ml to be then injected in a size exclusion chromatography (SEC). SEC was performed using an ÄKTA liquid chromatography system (Amersham Biosciences) and S75 10/300 GL (Tricorn) column (GE Healthcare) in 20 mM Tris-HCl pH 8.0 and 250 mM NaCl. After SEC, the fractions were pooled and concentrated and flash-frozen in liquid nitrogen and stored at −80 °C. For BLI experiments, ANTH and ENTH domains from Chaetomium thermophilum were produced using exactly the same protocol. Point-mutations were introduced by overlapping PCR with mutagenic primers (Supplementary Table 4).

ANTH-GFP and ENTH-GFP were cloned in pET11-SUMO vector and produced recombinantly under the same conditions. For purification, the SenP2 SUMO protease was used instead of TEV.

Protein concentrations were determined by absorbance measurements using NanoDrop with the extinction coefficients of the proteins calculated using ProtParam at Expasy.

Lipids: di-C$_8$-PI(4,5)P$_2$ was purchased from Echelon. DOPC (18:1 (Δ9-Cis) PC, Cat #850375), DOPS (18:1 PS, Cat #840035) and PIP$_2$ (08:0 PI(4,5)P2, Cat #85185 and Cat #840046) and 14:0 Liss Rhod PE (Cat # 810517) were purchased from Avanti Polar lipids.

**Thermal denaturation assays.** Proteins mixed to a final concentration of 30 μM in presence of 200 μM PIP$_2$ and incubated overnight at 4 °C were used to fill two standard grade NanoDSF capillaries (Nanotemper) and loaded into a Prometheus NT.48 device (Nanotemper) controlled by PR.ThermControl (version 2.1.2). Excitation power was pre-adjusted to get fluorescence readings between 2000 and 20000 RFU for fluorescence at 330 and 350 nm (F330 and F350, respectively), and samples heated from 20 °C to 90 °C with a slope of 1 °C/min. An apparent Tm was calculated from the inflection points of the fluorescence ratios (Ratio F350/F330) for ANTH and ENTH monomeric samples, where $\Delta T_m = T_m$ mutant − $T_m$ wild-type. To ensure we did not see any oligomerization interference effect potentially caused by domain instability, we discarded all mutants with a $\Delta T_m$ larger than 2 °C in our complex assembly study.

The scattering signal recorded (backscattering mode) was used as a stability reporter for the AENTH samples, where the mid aggregation point, $T_{agg}$, corresponds to the inflexion points in the scattering curves of the first transitions observed upon heating ($T_{agg}$ = mid-aggregation temperature obtained from scattering curves). $\Delta T_{agg}$ was calculated in the same way as done for $\Delta T_m$: $\Delta T_{agg} = T_{agg}$ mutant − $T_{agg}$ wild-type.

**Dynamic light scattering (DLS).** Measurements were performed using a DynaPro Nanostar device (Wyatt Technology Corporation) and data processed with Dynamics v.7 software. Proteins were mixed to a final concentration of 30 μM in presence of 200 μM PIP$_2$ and incubated overnight at 4 °C. Samples were centrifuged at 13,200 × *g* for 10 min at 4 °C prior to measurements using 4 μL plastic cuvettes (Wyatt Technology Corporation). The acquisition time was 5 s with a total of 30 acquisitions averaged. Measurements were performed at 25 °C.

**Batch SAXS measurements.** AENTH complexes were assembled at different protein concentrations (1–1.3 mg/mL) in the presence of 20 mM Tris-HCl pH 8.0,

250 mM NaCl and 0.2 mM PIP$_2$ and incubated overnight at 4 °C. Synchrotron radiation X-ray scattering data were collected (EMBL P12, PETRA III, DESY, Germany)[44,45] with a PILATUS 6 M pixel detector (DECTRIS, Switzerland) (20 × 0.05 s frames). Samples were measured through a capillary (20 °C). The sample-to-detector distance was 3.1 m, covering a range of momentum transfer $0.1 \leq q \leq 7$ nm$^{-1}$ ($q = 4\pi \sin\theta/\lambda$). Frame comparison showed no detectable radiation damage. Data were normalized, averaged, buffer subtracted, and placed on an absolute scale that is relative to water, according to standard procedures. All data manipulations were performed using PRIMUSqt and the ATSAS software package[46]. The forward scattering I(0) and radius of gyration, Rg were determined from Guinier analysis: $I(s) = I(0)\exp(-(sRg)^2/3)$). The indirect Fourier transform method was applied using the program GNOM[47] to obtain the distance distribution function p(r) and the maximum particle dimensions $D_{max}$. OLIGOMER[37] was used to calculate the proportion of the sample in the oligomeric and monomeric forms at equilibrium. As input, the pdb files for the monomers (PDB ID: 5OO7 AND 5ONF) and the structure of the 16-mer ($A_8E_8$) were used. In addition, to account for larger oligomeric structures, a 32-mer structure was created using SASREFMX assuming P2 symmetry[48].

**Stopped-flow time-resolved SAXS (SF-TR-SAXS) data collection and analysis**. A stopped-flow mixer (SFM 400, Bio-logic, Seyssinet-Pariset, France) equipped with a quartz capillary (0.8 mm inner diameter) was used as the sample delivery system at the SAXS beamline P12 (EMBL, PETRA III, DESY, Germany)[40,44]. The device was used to rapidly mix equimolar ratios of ANTH and ENTH domains at 2 mg/ml both in the presence of 200 μM PIP$_2$ with a dead time of 5 ms.

SAXS curves were recorded with an EIGER X 4 M detector at a distance of 3 m from the sample position using different time delays (0 ms, 400 ms, 800 ms, 1200 ms). These curves were collected in series spanning 600 ms and with a 200 ms overlap between each series. Each series contained spectra at 60-time points (10 ms spacing). Four data series with overlapping time points allowed SAXS curves were solvent subtracted (buffer and PIP$_2$ lipid) and used in the q-range between 0.01 and 2.9 nm$^{-1}$ for further analysis. SVD of the buffer-substracted SAXS curves was performed to obtain the number of components that contribute to the data using self-written Python code containing the modules Numpy[49], Scipy[50], and Scikit-learn[51]. SVD showed that the data can be described by 3 main components. (Supplementary Fig. 12). To minimize the possible impact of beam-induced radiation damage, the 10 first frames (timepoints) of each round were selected for further analysis. SVD using just 10 frames showed that the data could be described by two main components, whose contribution decreased and increased over time (monomers and complex formation, Supplementary Fig. 12). DAMMIX[41] was used to study the evolution from monomers to 16-mers an equilibrium in the TR-SF-SAXS data accounting for a possible intermediate and to obtain the ab initio model of the intermediate component.

**Grid preparation**. ANTH and ENTH at a concentration of 100 μM were pre-incubated in 200 μM PIP$_2$ in buffer containing 20 mM Tris pH 8.0, 250 mM NaCl and 1 mM DTT for 3 h at room temperature. Then, the solutions were mixed 1:1 to generate the AENTH complex and left on ice for at least 1 h. For cryo-EM grid preparation, Quantifoil 300 mesh Cu R 1.2/1.3 holey carbon grids were glow-discharged in a Cressington 208 carbon coater at 10 mA and 0.1 mbar air pressure for 30 s. The complex was diluted to 10 μM ANTH/ENTH (monomer) in buffer containing 200 μM PIP$_2$ and 3 μL was then applied to the grid and vitrified using a Vitrobot$^{TM}$ mark IV (FEI/Thermo Scientific) with a blot force of 6 and a blot time of 6 s. The relative humidity (RH) was ≥ 90% and temperature 5–6 °C. Liquid ethane was used as the cryogen.

**Data collection and processing**. Cryo-EM data were collected on a Titan Krios (FEI/Thermo Scientific) at the Astbury Biostructure Laboratory using a Falcon III direct electron detector operating in integrating mode. The main data acquisition parameters for the wildtype dataset ($A_8E_8$) are listed in Table 2. Processing of the $A_8E_8$ data was done using RELION 3[52] and cryoSPARC v2: Micrographs were corrected for beam-induced motion using MotionCor2[53] and the contrast transfer function (CTF) was estimated using Gctf[54], in RELION. Particles were picked initially using the general model in crYOLO[55] to generate 2D classes and a 3D reconstruction in RELION. The model was then trained to pick $A_8E_8$ particles. Using the trained model, 195,536 particles were picked from 7990 micrographs. A subset of these was used to generate an initial model and all particles were subjected to 3D classification to remove 'bad' particles. 96,664 particles were taken forward to refinement in C1 and D2 symmetry. Bayesian polishing[56] and beamtilt estimation were applied and a 2D classification step was performed on the polished particles to give a final selection of 79,414 particles, leading to a resolution of 4.1 Å (D2 symmetry). Non-uniform refinement in cryoSPARC v2[57] was used to further improve the resolution to 3.9 Å and the final reconstruction was sharpened with a B-factor of −200. An overview of the processing strategy is given in Supplementary Fig. 2.

Data collection parameters for the F5A/L12A/V13A mutant ($A_6E_6$) assembly are listed in Supplementary Table 2. Processing of the $A_6E_6$ data was done in RELION3[52]. Motion correction and CTF estimation were as for the $A_8E_8$ data. Using the general model in crYOLO, 142,399 particles were picked from 1,791

| Table 2 Cryo-EM data collection, refinement, and validation statistics of the AENTH 16-mer assembly. | |
|---|---|
| **Data collection and processing** | |
| Magnification | ×75,000 |
| Voltage (kV) | 300 |
| Electron exposure (e$^-$/Å$^2$) | 75.2 |
| Defocus range (μm) | −1 to −3 |
| Pixel size (Å) | 1.065 |
| Symmetry imposed | D$_2$ |
| Initial particle images (no.) | 195,536 |
| Final particle images (no.) | 79,414 |
| Map resolution (Å) | 3.9 |
| FSC threshold | 0.143 |
| Map resolution range (Å) | 3.7 to 5.7 |
| Refinement | |
| Initial model used | 5onf, 5oo7 |
| Model resolution (Å) | 3.9 |
| FSC threshold | 0.5 |
| Model resolution range (Å) | – |
| Map sharpening B factor (Å$^2$) | −200 |
| Model composition | |
| Non-hydrogen atoms | 26,192 |
| Protein residues | 3152 |
| Ligands | 20 |
| B factors (Å$^2$) | |
| Protein | 70 |
| Ligand | – |
| r.m.s. deviations | |
| Bond lengths (Å) | 0.012 |
| Bond angles (°) | 2.037 |
| Validation | |
| MolProbity score | 1.91 |
| Clashscore | 2.60 |
| Poor rotamers (%) | 3.78 |
| Ramachandran plot | |
| Favored (%) | 93.56 |
| Allowed (%) | 5.80 |
| Disallowed (%) | 0.64 |

micrographs. Those were subjected to two rounds of 2D classification, and a subset of 16,206 particles was selected. An initial model was generated from these particles, imposing C2 symmetry. The model was then refined with D3 symmetry, and particles were subjected to Bayesian polishing, CTF refinement, and beamtilt estimation, yielding a final reconstruction with 7.4 Å global resolution. An overview of the processing strategy is given in Supplementary Fig. 10.

**Model building**. As an initial model for the S. cerevisiae Sla2 ANTH domain, the Chaetomium thermophilum Sla2 ANTH domain crystal structure (PDB: 5oo7) was used. This, and the S. cerevisiae ENTH domain crystal structure (PDB: 5onf) were rigid-body docked into the cryo-EM map of the $A_8E_8$ complex using Chimera and manually adjusted in Coot[58]. The tetramer occupying the asymmetric unit was then iteratively refined in Coot and ISOLDE[59]. In total, 20 PIP$_2$ ligands were identified and placed in the model in Coot. Ligand coordinates and restraints were generated, symmetry applied and validation performed using tools in PHENIX 1.17[60].

For comparison with the previously published structure of ANTH-ENTH on lipid tubules, the atomic model for the $A_2E_2$ tetramer was docked into a subvolume of the EMD-2896 map and then flexibly fitted using adaptive distance restraints in ISOLDE.

Cryo-EM maps and atomic models were visualized using Chimera and ChimeraX[61,62].

**Native mass spectrometry**. Borosilicate nano-electrospray capillaries (Thermo Scientific) were prepared in-house using a P-97 micropipette puller (Sutter Instrument Co.) and coated with palladium/gold in a Polaron SC7620 sputter coater (Quorum Technologies). Native mass spectrometry measurements were done using an Orbitrap Q Exactive Plus UHMR (Thermo Scientific) operated in positive ion mode. Proteins were buffer-exchanged using one or two consecutive Zeba spin desalting columns (Thermo Scientific) into 300 mM ammonium acetate, 1 mM DTT, pH 8. 1 mg di-C$_8$-PI(4,5)P$_2$ (as sodium salt) dissolved in water was added to a mixture of ANTH and ENTH proteins (1:1 molar ratio) to form the

                                                                                                                 ARTICLE

complex at a final concentration of 10 μM monomeric proteins and 60 or 200 μM PIP₂. Instrument settings were 1.5–1.6 kV capillary voltage, −150 V in source trapping, HCD was off and the AGC target set to $3 \times 10^6$ with a maximum inject time of 300 ms. The trapping gas pressure (ratio) was 8, the mass range was 2000–20,000 $m/z$, and the resolution set to 6250. Raw data were processed and analyzed using UniDec[63].

**Biolayer interferometry.** Measurements were performed using an Octet RED96 instrument and Ni-NTA biosensors (ForteBio). Protein solutions were centrifuged for 10 min at 13,200×$g$ at 4 °C before the experiment to remove possible aggregates. Protein concentrations of these stock solutions were determined after centrifugation by the absorbance at 280 nm with a NanoDrop1000. Prior to the experiment, the biosensors were equilibrated in buffer I (50 mM Tris, pH 8; 125 mM NaCl and 0.05% BSA) for 10 min. Binding between His-tagged ANTH and ENTH from Chaetomium thermophilum was measured in the presence of 170 μM n-Dodecyl-β-D-Maltoside (DDM) and 50 μM PIP₂ in buffer I (buffer II). Prior attempts to use PIP₂ (225 μM) without DDM resulted in unspecific binding of ENTH to the biosensor. The experiments were performed at 25 °C with a shaking speed of 1000 rpm. An evaporation cover was used throughout the experiment.

Kinetic assays were performed in black, flat-bottom polypropylene 96 well plates (Greiner bio-one, item no. 655201) using 200 μl in each well. The kinetic assay consisted of 4 steps: 300 s equilibration in buffer I (baseline I), 300 s loading (3.75 μg/mL His-tagged ANTH in buffer I), 300 s equilibration in buffer II (baseline II), 600 s ENTH association (0.25 μM ENTH in buffer II), 1200 s dissociation in buffer II. To test for unspecific binding, a kinetic assay without loading His-tagged ANTH was used as control.

Data were visualized and analyzed with self-written Python scripts using the Python packages Numpy[49,64], Matplotlib[65] and Scipy[50]. Association and dissociation were fit with a biexponential function yielding a pair of two $k_{obs}$ and $k_{diss}$, each.

**Yeast growth assays.** Yeast strains MKY0764 (MATa, his3Δ200, leu2-3,112, ura3-52, lys2-801, sla2:natNT2) and MKY1421 (MATa, his3Δ200, leu2-3,112, ura3-52, lys2-801, tetENT2::HISMX6, ent1::hphNT1, tetR::LEU2) were transformed by pRS416-based centromeric plasmids expressing indicated variants of Sla2 ANTH domain (aa 1-289) and Ent1 ENTH domain (aa 1–154), respectively, generated by overlapping PCR with mutagenic primers. Growth of strains was followed by plating 10-fold serial dilutions of fresh cells on SD-Ura plates (containing 25 μg/ml of doxycycline in case of MKY1421) incubated at different temperatures for 2–4 days.

**Negative stain electron microscopy of LUVs.** Large unilamellar vesicles (LUVs) were prepared by extrusion. Lipids (76 mol% DOPC, 16 mol% DOPS, 8 mol% porcine brain PI(4,5)P₂ were mixed in 2:1 Chloroform/Methanol to give a total lipid weight of 1.8 mg. The solvent was evaporated under a stream of nitrogen gas and the lipid film dried in a vacuum. Lipids were resuspended in 183 μL buffer (20 mM Tris pH 8.0, 250 mM NaCl, and 1 mM DTT) to give a total lipid concentration of 10 mg/mL. After 5 freeze/thaw cycles, the suspension was passed through a 100 nm polycarbonate membrane 21 times in an extruder (Avanti Mini-Extruder).

To form AENTH tubules, ANTH, ENTH, and PIP₂-containing LUVs were mixed at final concentrations of 2 μM ANTH, 2 μM ENTH, and 0.1 mg/mL lipids in 20 mM Tris pH 8.0, 250 mM NaCl and 1 mM DTT. The mixture was incubated at room temperature for 90 s. Then, 3 μL of the sample were applied to a glow-discharged, carbon-coated copper grid. Grids were prepared in-house and glow discharged for 30 s in a Pelco easiGlow™ (Ted Pella) at 0.39 mbar and 12 mA. The sample was incubated on the grid for 30 s, then blotted and the grid was washed with H₂O three times and stained with 2% uranyl acetate twice for 60 s each. Excess stain was blotted off and the grid was dried. Grids were imaged using a Technai F20 microscope equipped with a Ceta (CMOS CCD) camera (FEI/Thermo Scientific).

**Fluorescence microscopy experiments.** GUVs were prepared following the classical electro formation protocol[66] Briefly, the lipid mixture (76 mol% DOPC, 16 mol% DOPS, 8 mol% 08:0 PI(4,5)P₂) dissolved in chloroform at concentration 1.25 mg/ml was mixed with 0.1% of 14:0 Liss Rhod PE and spread over two Pt nets and dried for 1 h in a vacuum. Then, a sucrose solution with the same osmolarity as the experimental buffer was added and a voltage of 220 V at an alternating frequency of 10 Hz was applied to the Pt nets using electrodes for 2 h. Later, the frequency was lowered to 2 Hz for 30 min. GUVs were allowed to settle for 30 min and observed using a Leica SP8 confocal microscope equipped with a white light laser and a ×63 NA 1.4 oil immersion objective.

To study the membrane deformation effects of ANTH and ENTH domains, both proteins were added at a final concentration of 1 μM to GUVs, with either ANTH or ENTH tagged with GFP. Z-stacks were acquired and representative equatorial sections are displayed in Fig. 6. Three independent experiments from 3 different GUV preparations were performed.

## Data availability

Data supporting the findings of this manuscript are available from the corresponding author upon reasonable request. A reporting summary for this article is available as a Supplementary Information file. Source data are provided with this paper. The atomic coordinates for the 16-mer AENTH complex model (PDBID 7A75) were deposited at the Protein Data Bank (https://www.ebi.ac.uk/pdbe/) PDB 7A75, the EM-map were deposited in EMBD (https://www.ebi.ac.uk/pdbe/emdb/) under the code EMD-11987 and EMD-11715.

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

## Acknowledgements

We would like to acknowledge the Sample Preparation and Characterization Facility at EMBL Hamburg. Synchrotron SAXS data were collected at beamline P12 operated by EMBL Hamburg at the PETRA III storage ring (DESY, Hamburg, Germany). We would like to thank beam scientists for the assistance in using the beamline. The work of M.A. and M.S. was supported by Deutsche Forschungsgemeinschaft (DFG) Research Grant SK 305/1-1. D.P.K. is a Ph.D. student on the Wellcome Trust 4-year Ph.D. programme in The Astbury Centre funded by The University of Leeds. The Titan Krios microscope was funded by the University of Leeds (UoL ABSL award) and Wellcome Trust (108466/Z/15/Z). The Orbitrap UHMR was funded by the University of Leeds and Wellcome Trust multi-user equipment grant 208385/Z/17/Z. The Technai F20 camera upgrade was funded by the Wellcome Trust (108466/Z/15/Z). J.L. is funded by the EMBL International Ph.D. programme. We would like to thank Benjamin Vollmer from the Grunewald lab for assistance with preparing GUVs.

## Author contributions

J.L. and K.V. produced proteins and samples for cryo-EM, SAXS, and native Mass spectrometry. J.L. performed biophysical experiments and interpreted EM Data. D.K. performed cryo-EM experiments and determined the AENTH structures together with S.M. D.K. performed mass spectrometry experiments and interpreted the data together with F.S. D.K. performed LUV negative stain EM experiments. J.L. performed fluorescence light microscopy experiments with GUVs supervised by R.T. M.A. and M.S. performed growth experiments in S. cerevisiae and interpreted data. M.A.S. and H.M. performed SAXS experiments and interpreted the data together with D.I.S. S.N. performed BLI experiments and SF-TR-SAXS data analysis with input from M.A.S. M.G.A. conceived and supervised the project. J.L. and M.G.A. wrote the manuscript with input from all authors.

## Funding

## Competing interests

The authors declare no competing interests.
