## [Peer Review File · Nature Communications]

Reviewer #1 (Remarks to the Author):

The manuscript titled "Structure of the endocytic adaptor complex reveals the basis for efficient membrane anchoring during clathrin-mediated endocytosis" is well executed and comprehensive. The authors report the oligomeric structure and interfaces of the adaptor complex AENTH of Sla2 and Ent1 in association with PIP2 using native MS and Cryo-EM, which compliment each other. It contributes to the understanding of protein-protein and protein-lipid interactions involved in the clathrin-mediated endocytosis.

Comments:

Please mention the molecular weight of the proteins in the manuscript. Please add that to the text. In figure 4(g), spectrum 1 is labeled as 'ANTH WT.' Should it be 'AENTH WT'? Also, there are some additional peaks which may be different numbers of PIP2 bound to the complex. Please label them clearly and mention in the text if they are nonspecific interactions.

Figure 4(g) - Please show the entire m/z range so that both monomeric and the multimeric species can be visible on the spectrum.

Figure 4(g) spectrum 5 - are they 6:6+PIP2 oligomers? If so, please state clearly in the text.

Moreover, specify the population of PIP2 adducts.

Figure 4g - Please change the labeling on each spectrum to show that it is an AENTH complex. (Follow the spectrum 2 style of labeling)

I recommend the paper to be accepted once the above concerns are addressed.

Reviewer #2 (Remarks to the Author):

You report a convincing cryoEM structure of a 16-mer complex of the membrane binding domains ENTH and ANTH bound to PIP2, which you say constitutes the anchor to the plasma membrane, as an important step in endocytosis. I understand that the structure and kinetic data on its assembly /disassembly is novel and sound, as far as I can judge (lacking the expertise in cryoEM).

As such, I am not sure, however, that the 16-mer complex by itself explains sufficiently how the anchoring works.

You mention force transmission, and that the complex 'facilitates the formation of larger assemblies which would contribute to membrane bending and remodeling'. But how does this work? How does the molecular structure reported help in this? What interactions with the membrane are relevant? Is the bound PIP2 still anchored in the membrane? From Fig.1, it is unclear where and how the authors envision the orientation with respect to the bilayer plane. Could you perform experiments of the complex interacting with model lipids to shed light on this, for example studies of the complex interacting with vesicles of different curvature?

A more detailed point to consider in a potential revision is the SAXS analysis, for example Fig.1.: (a) Experimental SAXS data of the 16-mer assembly (A8E8) in presence of 200 μ M PIP2: Please explain the model underlying the red curve. Does this correspond exactly to the A8E8 structure reconstructed from cryo-EM, or is this some empirical fit? Does the Cryo-EM structure also fit the distance distribution function of SAXS? Overall I find the description of the SAXS experiments

insufficient. How did structural models enter in the SVD and NMF approach in detail. Please also include the relevant references here.

Reviewer #3 (Remarks to the Author):

The authors present a structure of the membrane binding domains of two yeast proteins involved in clathrin-mediated endocytosis, Sla2 and Ent1, obtained using single particle cryoEM. At 3.9 Ang resolution, this map represents a substantial advance in detail over a previously published map of the Sla2 ANTH and Ent1 ENTH domains aligned in a helical array on lipid tubules. Intriguingly the authors find that these domains form a 16-mer composed of heterotetramers derived from both proteins that provides a new interpretation of the previous cryoEM map and reveals PIP2 binding sites in more detail. The authors also present functional studies that probe the interfaces between subunits. They show that the subunits can assemble rapidly that assembly and disassembly depend on concentration. This is a comprehensive study that uses a variety of experimental methods to characterise this complex and probe the functional significance of the new structural information. The results have implications for our understanding of the role of these assemblies in clathrin coat formation and provide a new interpretation likely to extend current thinking.

I would strongly support publication of this work subject to consideration of the concerns and suggestions listed below.

Structural model

The wwPDB validation report gives a relatively low score (below 50%) for Ramachandran and side chain outliers in the structure. Could the authors comment on whether these scores can be improved?

It took a while to discover that the numbering in the PDB file was 5 residues greater than the numbering used in the text. Could this be reconciled or explained in the text and PDB entry? Also, it would be very helpful if the chains could be labelled such that the ENTH and ANTH domains can be distinguished easily when examining the structure.

Manuscript

Page 9 lines 216-220 The authors appear to state that they have found the true physiologically relevant form of the ANTH/ENTH interaction. While this seems to be a reasonable new interpretation of the earlier helical map, the wording here seems too strong for the evidence provided. Please rephrase or provide additional justification.

Page 9 line 227-229 – Could the lysine patch on the ANTH domain close to the membrane be illustrated, e.g. on figure S9?

Page 9 lines 232-234 – The difference between the open and closed forms identified is interesting. Could a direct comparison of the two forms be shown e.g. on figure S9b or perhaps more prominently on a main figure?

Page 11 279-183 – The interpretation of the time-resolved SAXS experiments does not appear to do

the technique justice. Was structural modelling attempted to gain further insights into the nature of the changes observed? Currently the only interpretation given is that there was a rapid increase in complex formation, which could arguably have been discovered using a much less sophisticated technique. If structural modelling was attempted but was unsuccessful or not appropriate in this application it would be helpful if the authors could comment on this.

Page 23 lines 544-545 – What was the reason for incubating the sample for an hour? The SAXS experiments indicated that complex formation was much more rapid than this. Could the structure have altered during that time to form a stable but less functionally relevant assembly?

Minor points

Page 3 line 61 - 'These adaptors have the topology of elongated knot and string proteins' – This needs some explanation and/or a reference.

Page 5 line 107 567 - When the global resolution of the map is stated in the text it would be helpful to also state the criteria used. I appreciate this is in the table but I think it's still important for clarity.

Page 6 line 149 – Figure S6a is referred to but the melting temperature experiments are not commented on.

Legend to Figure 2 – 'The density corresponding to the polar head of the PIP2 is shown in mesh.' I can't see any density represented as mesh on this figure. Could this be clarified?

Typos

Page 2 line 52 –'an' is needed between 'not' and 'absolute'

Page 9 Line 227 After 'similarly', 'the' is lacking a t

Comments to the reviewers and replies

REVIEWER COMMENTS

Reviewer #1 (Remarks to the Author):

The manuscript titled “Structure of the endocytic adaptor complex reveals the basis for efficient membrane anchoring during clathrin-mediated endocytosis” is well executed and comprehensive. The authors report the oligomeric structure and interfaces of the adaptor complex AENTH of Sla2 and Ent1 in association with PIP₂ using native MS and Cryo-EM, which compliment each other. It contributes to the understanding of protein-protein and protein-lipid interactions involved in the clathrin-mediated endocytosis.

Comments:

1) Please mention the molecular weight of the proteins in the manuscript. Please add that to the text.

Response: We have added a table (**Table 1**) indicating the experimental and theoretical masses for ANTH, ENTH, PIP₂ lipids bound and complexes.

The following text was added on page 4, line 100: “The ANTH and ENTH domains from Sla2 and Ent1 (33.2 and 18.9 kDa, respectively, see **Table 1**)...”

2) In figure 4(g), spectrum 1 is labeled as ‘ANTH WT.’ Should it be ‘AENTH WT’? Also, there are some additional peaks which may be different numbers of PIP₂ bound to the complex. Please label them clearly and mention in the text if they are nonspecific interactions.

Response: The labels on Figure 4(g) have been corrected and the number of PIP₂ molecules described in **Table 1**. Furthermore, Supplementary **Fig 7** shows in detail that the different peaks correspond to different PIP₂ molecules bound to the A₈E₈ complex.

3) Figure 4(g) - Please show the entire m/z range so that both monomeric and the multimeric species can be visible on the spectrum.

Response: We have added a supplementary figure (Supplementary **Fig.7**) showing the whole m/z range spectrum collected for the ANTH WT + ENTH WT sample. The presence of free ANTH and ENTH is observed at lower m/z values while the complex appears at around 10000 m/z

(Supplementary **Fig. 7a**). As suggested by the reviewer, those additional peaks corresponding to different numbers of PIP₂ (22, 23 and 24 are clearly distinguishable) bound to the complex (highest intensity +41 charge state of the 8:8 ANTH/ENTH complex) are now shown in Supplementary **Fig. 7c**. A close up of the spectrum at lower m/z values where ANTH and ENTH are bound to 1, 2 and 3 PIP₂ molecules is shown in Supplementary **Fig. 7d**. For the ANTH domain, species bound to 1, 2 and 3 Na⁺ molecules (coming from the PIP₂ sample) were also detected (see Supplementary **Fig. 7e** and **Materials and Methods**).

The main text of the manuscript has been modified and the following added:

-Page 7, line 171: “Native MS allows the identification of charge state distributions corresponding to A₈E₈ complexes at higher m/z, with 22-24 PIP₂ molecules bound (Supplementary **Fig. 7**).”

-Page 10, line 250: “In addition to the 20 PIP₂ lipids described in the entire 16-mer complex, up to 24 PIP₂ molecules were detected by native MS indicating the presence of four additional bound lipid molecules that could not be resolved in our structure (Supplementary **Fig. 7c**).”

4) Figure 4(g) spectrum 5 - are they 6:6+PIP₂ oligomers? If so, please state clearly in the text. Moreover, specify the population of PIP₂ adducts.

Response: As the reviewer points out, the ANTH R3A/I4A/D37R/H38A mutant spectrum corresponds to a 12-mer (A₆E₆) complex. The ANTH R3A/I4A/D37R/H38A masses and charge states, as well as the number of PIP₂ bound are content in **Table 1**. We have relabeled the spectrum in **Fig. 4g**.

The following text has been added to the main text of the manuscript:

-Page 9, line 224: “Interestingly, native MS for the R3A/I4A/D37R/H38A mutant showed a shift in the signal of the complexes obtained to lower m/z, corresponding to 12-mers (**Fig. 4g** and **Table 1**), in agreement with the DLS data. Assemblies of 12-mers have been previously reported as lower abundance species^{19,36} and are formed by 6 ANTH and 6 ENTH molecules (termed A₆E₆). However, mutation of the ANTH-ANTH interface did not cause growth defect phenotype *in vivo* (Supplementary **Fig. 9c**).”

-Page 12, line 307: “In case of ANTH R3A/I4A/D37R/H38A, the typical 16-mer AENTH complex was not observed for this mutant, but a 12mer assembly (**Fig. 4g** and **Table 1**) previously described corresponding to 6 ANTH and 6 ENTH molecules (termed A₆E₆)^{19,36}, was sufficient to introduce membrane reshaping *in vitro*.”

5) Figure 4g - Please change the labeling on each spectrum to show that it is an AENTH complex. Follow the spectrum 2 style of labeling.

Response: Figure 4g has been relabeled.

6) I recommend the paper to be accepted once the above concerns are addressed.

Reviewer #2 (Remarks to the Author):

You report a convincing cryoEM structure of a 16-mer complex of the membrane binding domains ENTH and ANTH bound to PIP2, which you say constitutes the anchor to the plasma membrane, as an important step in endocytosis. I understand that the structure and kinetic data on its assembly /disassembly is novel and sound, as far as I can judge (lacking the expertise in cryoEM).

1) “As such, I am not sure, however, that the 16-mer complex by itself explains sufficiently how the anchoring works”.

Response: We would like to point out that we are proposing that the anchoring structure is a tetramer that can further associate into bigger assemblies. We have presented the first high-resolution structure of the AENTH complex bound to lipids. The structure was solved for a 16-mer assembly composed of four AENTH tetramers. We have shown how mutations on the tetrameric interfaces disrupt the complex formation (no tetramers detected, therefore no bigger assemblies) *in vitro* and have associated growth defect phenotypes *in vivo*. We have now performed additional experiments on GUVs and LUVs to show the crucial role of the reported tetrameric AENTH structure in membrane remodeling. We now thoroughly revised the relevant parts of the manuscript:

2) “You mention **force transmission**, and that the complex '**facilitates the formation of larger assemblies which would contribute to membrane bending and remodeling**.”

Response: We have modified the text to better stress the role of the AENTH tetramer in membrane anchoring (required for force transmission) and improved the section for the formation of larger assemblies. Regarding its role in membrane remodeling we have performed two new experiments.

New experiments:

Following the reviewer’s comment, new experiments performed on GUVs and LUVs (**Fig. 6 and 7**) show how ANTH and ENTH mutants that are not able to form the AENTH tetramer fail to contribute to membrane remodeling using *in-vitro* model membranes.

2) “But how does this work ? How does the molecular structure reported help in this?”

Response: We have repurposed a main figure (**Fig. 5**) to depict the involved protein-lipid interactions and the role of the tetramer in membrane anchoring.

3) What interactions with the membrane are relevant ?

Response: As requested by the reviewer we have highlighted the regions involved in lipid clamping on **Fig. 5e**.

4) Is the bound PIP₂ still anchored in the membrane ?

Response: Our experiments using model membranes (liposomes and GUVs) indicate that PIP₂ needs to be present in the lipid bilayer to see protein binding. Please see response to point 6 below. **Fig. 6a** and **6b** indicate that PIP₂ needs to be present in the bilayer for deformation to occur. Colocalization of the GFP signal (protein) and Rhodamine labelled-PE (membrane) in our confocal fluorescence microscopy experiments further confirms that the assembly of the domains takes place in the membrane.

5) From Fig.1, it is unclear where and how the authors envision the orientation with respect to the bilayer plane.

Response: This has been addressed in the new **Fig. 5**. The model of a tetramer interacting with the membrane is shown in panels **c to e** and the membrane plane indicated with a dashed line.

6) Could you perform experiments of the complex interacting with model lipids to shed light on this, for example studies of the complex interacting with vesicles of different curvature?

Response: Following the reviewer's advice, we have performed experiments with model membranes. We have used liposomes/Large Unilamellar Vesicles (LUVs) as a high curvature model and Giant Unilamellar Vesicles (GUVs) as lower curvature model.

New figures (**Fig. 6 and 7**) have been added to the manuscript. When ANTH and ENTH domains are combined in the presence of GUVs containing PIP₂ the AENTH complex reshapes GUVs into tubular structures, as previously reported by Skruzny et al., (2015). We produced ANTH-GFP or ENTH-GFP, mixed them with GUVs with and without PIP₂ and imaged them using confocal fluorescence microscopy. ANTH-GFP or ENTH-GFP by themselves did not

cause remodeling effect of GUVs, while when both proteins were together hairy structures protruding from the vesicles could be visualized. Importantly, ANTH and ENTH tetrameric mutants had different membrane deformation capabilities. Those mutants shown to be unable to form the AENTH complex have failed to form the hairy structures observed for the wild-type domains.

Similar results were obtained for LUVs containing PIP₂ observed by negative staining EM. ANTH does not seem to deform LUVs notably, in agreement with literature (Ford et al. 2001). ENTH causes LUVs to become aggregated. Both domains together then lead to clear LUVs tubulation. As for GUVs, those mutants that cannot assemble a functional AENTH complex were unable to tubulate LUVs. The conclusion from these experiments using membrane models is that only ANTH and ENTH proteins which can assemble into tetrameric (and therefore bigger assemblies) are able to remodel membranes.

7) “A more detailed point to consider in a potential revision is the SAXS analysis,

7.1. for example Fig.1.: (a) Experimental SAXS data of the 16-mer assembly (A₈E₈) in presence of 200 μM PIP₂: Please explain the model underlying the red curve”.

Response: We acknowledge that SAXS data was presented with a limited explanation and analysis in the previous version of our manuscript. A new version of Supplementary **Fig. 1** including SAXS curves at different concentrations has been created addressing this point raised by the reviewer. (see Supplementary **Fig. 1a**). So indeed, the curves fitted to the experimental SAXS data are derived from mixtures in equilibrium, not from the theoretical scattering curve of the 16-mer (see below).

7.2. “Does this correspond exactly to the A₈E₈ structure reconstructed from cryo-EM, or is this some empirical fit? “

Response: No, it does not correspond to the 16-mer A₈E₈ structure scattering curve. It is a fit of a mixture of oligomers based on the combination of the experimental data available: monomers, 16-mer and 32-mer complexes existing in solution as have been previously described by Heidemann *et al.* 2020.

7.3) “Does the Cryo-EM structure also fit the distance distribution function of SAXS?”

Response: No, the 16-mer (A₈E₈) complex alone does not fit the data because the system consists of a mixture in solution (see previous response). The 16-mer overall dimensions agree with the

distance distribution function, however the shape of the ab-initio models generated from this model does differ from the one of the cryo-EM structure. The reason for this is the heterogeneity present in solution, which is “filtered out” during particle selection and alignment during cryo-EM data processing to achieve higher resolution of a subset of the particles embedded in the ice.

8) “Overall I find the description of the SAXS experiments insufficient. How did structural models enter in the SVD and NMF approach in detail. Please also include the relevant references here”.

Response: As suggested by the reviewer, we have now extended the analysis of the time resolved SAXS data (see **Fig. 8** and Supplementary **Fig. 11**) and re-written the **Kinetics of the AENTH complex assembly** and **Material and Methods** sections.

Reviewer #3 (Remarks to the Author):

The authors present a structure of the membrane binding domains of two yeast proteins involved in clathrin-mediated endocytosis, Sla2 and Ent1, obtained using single particle cryoEM. At 3.9 Ang resolution, this map represents a substantial advance in detail over a previously published map of the Sla2 ANTH and Ent1 ENTH domains aligned in a helical array on lipid tubules. Intriguingly the authors find that these domains form a 16-mer composed of heterotetramers derived from both proteins that provides a new interpretation of the previous cryoEM map and reveals PIP2 binding sites in more detail. The authors also present functional studies that probe the interfaces between subunits. They show that the subunits can assemble rapidly that assembly and disassembly depend on concentration. This is a comprehensive study that uses a variety of experimental methods to characterise this complex and probe the functional significance of the new structural information. The results have implications for our understanding of the role of these assemblies in clathrin coat formation and provide a new interpretation likely to extend current thinking.

I would strongly support publication of this work subject to consideration of the concerns and suggestions listed below.

Structural model

1) The wwPDB validation report gives a relatively low score (below 50%) for Ramachandran and side chain outliers in the structure. Could the authors comment on whether these scores can be improved?

Response: We have conducted extensive model building and refinement in COOT and ISOLDE and have attempted to improve Ramachandran and side chain outliers, but were only able to do so at the expense of an increased clash-score. Although the scores are below 50% the majority of the model can reliably be built with the problematic regions being those with relatively lower resolution at the periphery of the complex, which proved challenging to build.

2) It took a while to discover that the numbering in the PDB file was 5 residues greater than the numbering used in the text. Could this be reconciled or explained in the text and PDB entry?

Response: We apologise for the confusion caused and have now corrected the numbering in the PDB file to match the labels/numbering in the main text.

3) Also, it would be very helpful if the chains could be labelled such that the ENTH and ANTH domains can be distinguished easily when examining the structure.

Response: We have added appropriate labels for all chains. ENTH chains are now labelled as “ENTH domain of epsin Ent1” and ANTH chains are labelled as “ANTH domain of Sla2”.

Manuscript

1) “Page 9 lines 216-220 The authors appear to state that they have found the true physiologically relevant form of the ANTH/ENTH interaction. While this seems to be a reasonable new interpretation of the earlier helical map, the wording here seems too strong for the evidence provided. Please rephrase or provide additional justification”.

Response: The text has been re-phrased and additional experiments showing the remodeling properties of the AENTH complex on GUVs and LUVs have been performed.

2) Page 9 line 227-229 – Could the lysine patch on the ANTH domain close to the membrane be illustrated, e.g. on figure S9?

Response: The previous Supplementary **Fig. 9** has now been modified into main **Fig. 5** (panels b to e). A model illustrating the residues interacting with PIP₂ in the membrane are shown in yellow (see **Fig. 5e**). It is important to highlight that sidechain positions cannot be reliably determined at 13 Å resolution.

3) Page 9 lines 232-234 – The difference between the open and closed forms identified is interesting. Could a direct comparison of the two forms be shown e.g. on figure S9b or perhaps more prominently on a main figure?

Response: As suggested by the reviewer this has now been added as a main figure (**Fig. 5**) and a supplementary movie is available to see the closed and open conformations. A description of this observations has been updated in the main text. See page10-11, lines 256-274

4) “Page 11 279-183 – The interpretation of the time-resolved SAXS experiments does not appear to do the technique justice. Was structural modelling attempted to gain further insights into the nature of the changes observed?

Currently the only interpretation given is that there was a rapid increase in complex formation, which could arguably have been discovered using a much less sophisticated technique. **If structural modelling was attempted but was unsuccessful or not appropriate in this application it would be helpful if the authors could comment on this”.**

Response: We have now extended the analysis of the time resolved SAXS data (see **Fig. 8** and Supplementary **Fig. 11**) and re-written the **Kinetics of the AENTH complex assembly** and **Material and Methods** sections. As indicated in the re-analysis of the SAXS batch

measurements, the system in equilibrium consists already of a mixture of monomers and oligomers in solution. Following each single species evolution in our time-resolved SAXS datasets is unfortunately not possible. The different species and the limited quality of the SAXS curves makes finer structural modelling extremely challenging. For this reason, DAMMIX was used to treat all the intermediate species as an average of the intermediate states between monomers and complexes. The dimensions of the tetramer and of the DAMMIX *ab initio* model are rather similar, which could indicate that the unknown component of the SAXS data could be tetramers forming in solution that later on assemble into larger oligomeric states (see Fig. 8). However, we are aware that this highly speculative statement will need to be supported in future time-resolved techniques about AENTH assemblies.

5) “Page 23 lines 544-545 – What was the reason for incubating the sample for an hour? The SAXS experiments indicated that complex formation was much more rapid than this. Could the structure have altered during that time to form a stable but less functionally relevant assembly?”

Response: We agree with the reviewer that the equilibrium state (one hour incubation) differs from the fast assembly reported after 1200 ms. This has been now reflected in the manuscript (page 13, line 337. The 1 hour incubation time was chosen and kept constant throughout all experiments and helped in to obtain high quality grids and gave us time to perform all biophysical measurements (native MS, DLS and nanoDSF). We believe that one-hour incubation represents a sample in equilibrium which contains a mixture of 16mers and 32mers (see Supplementary Fig. 1). We have not observed any aggregation or DSF destabilization of the sample after the incubation time.

Minor points

1) Page 3 line 61 - ‘These adaptors have the topology of elongated knot and string proteins’ – This needs some explanation and/or a reference.

Response: The following references are added to the main text:

-Busch, D. J. et al. Intrinsically disordered proteins drive membrane curvature. Nat. Commun. 6, 1–11 (2015).

-Engqvist-Goldstein, Å. E. Y. et al. The actin-binding protein Hip1R associates with clathrin during early stages of endocytosis and promotes clathrin assembly *in vitro*. *J. Cell Biol.* 154, 1209–1223 (2001).

-Holkar, S. S., Kamerkar, S. C. & Pucadyil, T. J. Spatial control of epsin-induced clathrin assembly by membrane curvature. *J. Biol. Chem.* 290, 14267–14276 (2015).

Particularly, Engqvist-goldstein shows in Figure 6 the elongated topology of Hip1R (Sla2 homologue) by electron microscopy.

2) Page 5 line 107 567 - When the global resolution of the map is stated in the text it would be helpful to also state the criteria used. I appreciate this is in the table but I think it's still important for clarity.

Response: The text has been modified as follows:

Page 5, line 117: “The global resolution of the final EM map is 3.9 Å (gold standard Fourier shell correlation threshold 0.143) with...”

3) Page 6 line 149 – Figure S6a is referred to but the melting temperature experiments are not commented on.

Response: nanoDSF experiments on the mutants are shown as a control of whether the mutation itself is destabilizing the fold of the domain. We are now commenting Supplementary **Fig. 6a-c** in different parts of the manuscript:

Page 6 line 156: To test the relevance of this interface for complex formation, point mutations were introduced in residues in the ANTH and ENTH domains present in this interface and the stability of recombinant proteins was analyzed by nanoDSF (Supplementary **Fig. 6a**).

Page 7 line 179: ENTH F108A also introduced a growth defect phenotype *in vivo* (**Fig. 4i**), but our *in vitro* data showed that the stability of the protein is compromised by this mutation (Supplementary **Fig. 6a**).

Page 8 line 188: A second interface in the AENTH tetramer involves residues K10, K13 and K14 of ANTH (**Fig. 4b** and Supplementary **Fig. 6b**).

Page 9 line 220: Finally, ANTH-ANTH interface mutants (Supplementary **Fig. 6c**) did not show a large destabilization effect over the complex *in vitro* with...

4) Legend to Figure 2 – ‘The density corresponding to the polar head of

the PIP2 is shown in mesh.' I can't see any density represented as mesh on this figure. Could this be clarified?

Response: This has been modified. The density from the cryo-EM map is shown in transparent surface representation.

Typos

5) Page 2 line 52 –'an' is needed between 'not' and 'absolute'
-corrected.

6) Page 9 Line 227 After 'similarly', 'the' is lacking a t
-corrected.

Reviewer #1 (Remarks to the Author):

The authors have answered all my queries satisfactorily. Hence, I recommend that the manuscript be accepted for publication.

Reviewer #2 (Remarks to the Author):

I find your reply and revision convincing and am pleased to recommend publication.

Reviewer #3 (Remarks to the Author):

The revised manuscript is much improved and addresses all my prior concerns. The new movie showing open and closed states is particularly helpful and the further experiments have added value to this already comprehensive study. I am happy to recommend publication.